nanotechnology/quantum physics

field electron emission, current–voltage data interpretation, field emitter characterization, Murphy–Good theory, Fowler–Nordheim plots, Murphy–Good plots

**Author for correspondence:**
Richard G. Forbes
e-mail: r.forbes@trinity.cantab.net

# The Murphy–Good plot: a better method of analysing field emission data

## Richard G. Forbes

Advanced Technology Institute & Department of Electrical and Electronic Engineering, University of Surrey, Guildford, Surrey GU2 7XH, UK

 RGF, 0000-0002-8621-3298

Measured field electron emission (FE) current–voltage $I_m(V_m)$ data are traditionally analysed via Fowler–Nordheim (FN) plots, as $\ln\{I_m/V_m^2\}$ versus $1/V_m$. These have been used since 1929, because in 1928 FN predicted they would be linear. In the 1950s, a mistake in FN's thinking was found. Corrected theory by Murphy and Good (MG) made theoretical FN plots slightly curved. This causes difficulties when attempting to extract precise values of emission characterization parameters from straight lines fitted to experimental FN plots. Improved mathematical understanding, from 2006 onwards, has now enabled a new FE data-plot form, the 'Murphy–Good plot'. This plot has the form $\ln\{I_m/V_m^{(2-\eta/6)}\}$ versus $1/V_m$, where $\eta \cong 9.836239$ $(\mathrm{eV}/\phi)^{1/2}$ and $\phi$ is the local work function. Modern (twenty-first century) MG theory predicts that a theoretical MG plot should be 'almost exactly' straight. This makes precise extraction of well-defined characterization parameters from ideal $I_m(V_m)$ data much easier. This article gives the theory needed to extract characterization parameters from MG plots, setting it within the framework of wider difficulties in interpreting FE $I_m(V_m)$ data (among them, use of 'smooth planar emitter methodology'). Careful use of MG plots could also help remedy other problems in FE technological literature. It is suggested that MG plots should replace FN plots.

## 1. Background

Field electron emission (FE) occurs in many technological contexts, especially electron sources and electrical breakdown. A need exists for effective analysis of measured FE current–voltage [$I_m(V_m)$] data, to extract emission characterization parameters. These include: parameters that connect field to voltage; the field enhancement factors (FEFs) often used to characterize large-area field-electron emitters (LAFEs); and parameters relating to emission area and area efficiency (the latter being a measure of what fraction of emitter area is emitting significantly). This article proposes a simple new method for FE $I_m(V_m)$ data analysis, and

urges its widespread adoption. This method, the Murphy–Good (MG) plot, is in principle more precise than the traditional Fowler–Nordheim (FN) plot. The article represents one individual small part of a much wider project, partly outlined elsewhere [1], that aims to put FE theory onto a better scientific basis, by (among other things) improving the precision of interpretation of FE experimental data.

To develop MG-plot theory efficiently, some preliminary discussion and refinement of traditional FN theory is needed. FN plots were introduced by Stern *et al.* [2] in 1929. They have the form $\ln\{I_m/V_m^2\}$ versus $1/V_m$ (or equivalent using other physical variables), and are used because the original 1928 FN equation [3] implied that FN plots of experimental data should be straight lines, with characterization data derivable from the slope and intercept.

However, in 1953, Burgess, Kroemer & Houston (BKH) [4] found a mathematical mistake in 1928 theoretical work by Nordheim [5], and a related physical mistake in FN's thinking. FN had assumed [3] that image-force rounding could be disregarded, and treated the electron tunnelling barrier as exactly triangular (ET). BKH showed that rounding was much more important than FN had thought and Nordheim had calculated, and that (for emitters modelled as planar) it is *necessary* to base analyses on planar image-rounded barriers (often now called 'Schottky–Nordheim' (SN) barriers). Corrected analysis inserted a 'barrier form correction factor' into the exponent of the original 1928 FN equation, and led to much higher tunnelling probabilities (typically, by a factor between 250 and 500 [1]). This correction factor is generated by an appropriate value of a special mathematical function (SMF) $v(x)$ now known [6] to be a special solution of the Gauss hypergeometric differential equation. The *Gauss variable* $x$ is the independent variable in this equation.

The Nordheim parameter $y$ used in older FE discussions is given by $y = +x^{1/2}$, but its use *in mathematical contexts* can now be recognized as illogical—when a function is the solution of a differential equation, mathematics does not normally represent it as a function of the square root of the independent variable in the equation. The use of $y$ (rather than $x$ [$=y^2$]) in FE literature is due to an unfortunate arbitrary choice (separate from the above mistake) made by Nordheim in his 1928 paper. Although $y$ is useful as a modelling parameter in some theoretical discussions, hindsight indicates that choosing to use $x$ [$=y^2$] in 1928 would have proved better mathematics (and better for discussing FE $I_m(V_m)$ data).

In 1956, Murphy and Good (MG) [7] used the BKH results to develop a revised FE equation. (See [8] for a treatment that uses the modern 'International System of Quantities' (ISQ) [9]). The zero-temperature version of their equation is called here the *Murphy–Good (MG) FE equation.* This equation is an adequate approximation at room temperature.

The MG FE equation gives the local emission current density (LECD) $J_L^{MG}$ in terms of the local work function $\phi$ and local barrier field $F_L$. It is clearest to start from the linked form

$$J_L^{MG} = t_F^{-2} J_{kL}^{SN}, \tag{1.1a}$$

$$J_{kL}^{SN} = a\phi^{-1}F^2\exp\left[\frac{-v_F b\phi^{3/2}}{F_L}\right], \tag{1.1b}$$

where $a$ [$\cong 1.541434\ \mu A\ eV\ V^{-2}$] and $b$ [$\cong 6.830890\ eV^{-3/2}\ V\ nm^{-1}$] are universal constants [10], often called the *first* and *second FN constants*, $v_F$ is the value of $v(x)$ that applies to the SN barrier defined by $\phi$ and $F_L$, and $t_F$ is the corresponding value of an SMF $t(x)$ defined by

$$t(x) = v(x) - \left(\frac{4}{3}\right)x\frac{dv}{dx}. \tag{1.2}$$

$J_{kL}^{SN}$ is called the *kernel current density for the SN barrier*, and can be evaluated *precisely* when $\phi$ and $F_L$ are known.

The correction factor $v_F$ is field-dependent (see below). This causes theoretical FN plots predicted by the MG FE equation to be slightly curved, rather than straight. This, in turn, causes very significant problems of detail and the need for related procedures, when attempts are made to give well-defined *precise* meanings to the slope and intercept of the straight line fitted to an FN plot of experimental data. These interpretation procedures involve correct choice of fitting point [11,12] and application of a chord correction [12]. This article shows how to eliminate these particular problems, by finding a plot form that the MG FE equation predicts to be 'almost exactly' linear.

An *ideal* FE device/system is one in which the measured current–voltage $I_m(V_m)$ characteristics are determined only by unchanging system geometry and by the emission process (see [13], and below). If curvature in an FN plot taken from an ideal FE device/system is due to physical reasons (such as small apex radius of curvature, or—with a LAFE—statistical variations in the characteristics of

individual emitters), then use of an MG plot will not be able to straighten out this kind of curvature, though it should be a useful step forwards.

# 2. Some general issues affecting field emission current–voltage data analysis

In fact, three other major problems affect both FN-plot and MG-plot interpretation, and need discussion. The FN and MG derivations disregard the existence of atoms and model the emitter surface as smooth, planar and structureless. This *smooth planar emitter methodology* is unrealistic, but creating reliably better theory is very difficult, although there are some atomic-level treatments, e.g. [14,15]. When applying a current–density equation to real emitters, this weakness can be explicitly formalized, as follows.

To recognize both this difficulty and all other factors omitted in deriving equation (1.1*b*), the present author has replaced $t_F^{-2}$ in equation (1.1*a*) by an 'uncertainty factor' $\lambda$ of unknown functional behaviour (see [16] for recent discussion). I now write $J_L^{EMG} = \lambda J_{kL}^{SN}$, and call the revised equation (and variants using other physical variables) the *Extended Murphy–Good (EMG) FE equation*. My current thinking [17] is that (for a SN barrier) $\lambda$ varies with relevant parameters (in particular, local field) and most probably lies somewhere in the range $0.005 < \lambda < 14$ (though it could turn out, when further atomic-level treatments are available, that the figure 0.005 has been pessimistic and unnecessarily low [17]).

In principle, the related total emission current ($I_e^{EMG}$) is found by integrating $J_L^{EMG}$ over the emitter surface and writing the result as first shown below, where $A_n^{EMG}$ is the *notional emission area* (as derived using the EMG equation),

$$I_e^{EMG}(F_C) = \int J_L^{EMG} dA = A_n^{EMG} J_C^{EMG} = A_n^{EMG} \lambda J_{kC}^{SN} \equiv A_f^{SN} J_{kC}^{SN}. \tag{2.1}$$

The subscript 'C' denotes characteristic values taken at some characteristic location on the emitter surface (in modelling, nearly always the emitter apex).

The second form follows from $J_C^{EMG} = \lambda J_{kC}^{SN}$. Often, $\lambda$ and $A_n^{EMG}$ are both unknown. Equations with two unknown parameters are inconvenient, so these are combined into a single parameter $A_f^{SN}[\equiv \lambda A_n^{EMG}]$ called the *formal emission area for the SN barrier*.

Combining these various relations, and assuming that measured current $I_m$ equals emission current $I_e^{EMG}$, yields the following EMG-theory equation:

$$I_m(F_C) = A_f^{SN} J_{kC}^{SN} = A_f^{SN} a \phi^{-1} F_C^2 \exp\left[\frac{-v_F b \phi^{3/2}}{F_C}\right]. \tag{2.2}$$

Although this is not explicitly shown, it needs to be understood that the values of $\phi$, $v_F$, $\lambda$, $A_n$, and $A_f$ depend on the choice of location 'C'.

When applying this equation to experiments, and 'thinking backwards', $I_m(F_C)$ is a measured quantity, and $J_{kC}^{SN}$ can be calculated precisely (when $\phi$ and $F_C$ are known). Thus, the *extracted* parameter $\{A_f^{SN}\}^{extr}[= I_m(F_C)/J_{kC}^{SN}]$ is, in principle, a well-defined parameter that depends on the barrier form, but not on $\lambda$: thus, the symbol $\{A_f^{SN}\}^{extr}$ carries the barrier label, rather than an equation label.

In practice, it is nearly always the *formal* area that is initially extracted from an FN or MG plot. Issues of how formal area relates to the notional area in some specific emission equation, or to geometrical quantities relating closely to real emitters, are matters for separate discussion, outside the scope of this paper. This paper is primarily about the extraction of precise values for formal area $A_f^{SN}$.

A second major problem lies in determining the relationship between the characteristic barrier field $F_C$ and the measured voltage $V_m$. I now prefer to write

$$F_C = V_m / \zeta_C, \tag{2.3}$$

where $\zeta_C$ is the *characteristic voltage conversion length (VCL)*, for location 'C'. Except in special geometries, $\zeta_C$ is not a physical length. Rather, $\zeta_C$ is a system characterization parameter: low VCL means the emitter 'turns on' at a relatively low voltage $V_m$.

So-called *ideal* FE devices/systems have $I_m(V_m)$ characteristics determined *only* by the system geometry (which must be unchanging) and by the emission process, with no 'complications' (see below). For ideal devices/systems, the VCL $\zeta_C$ is *constant*, and related characterization parameters (such as characteristic FEFs) can be derived from extracted $\zeta_C$-values (see below, and also [16]).

However, real FE devices/systems may have 'complications', such as (among others) leakage current, series resistance in the measuring circuit, current dependence in FEFs, and space-charge effects.

These may cause 'non-ideality' whereby $\zeta_C$ ceases to be constant but becomes dependent on voltage and/or current. In turn, this may modify the FN or MG plot slope or cause plot nonlinearity. In such cases, conventional FN-plot analysis may generate spurious results for characterization parameters [13,16]. This will also be true for MG-plot analysis.

Additional research is urgently needed on how to analyse and model the FE $I_m(V_m)$ characteristics of non-ideal devices/systems, but it will probably be many years before comprehensive theory exists. Hence, at present, FN and MG plots provide adequate emission characterization *only* for ideal devices/systems. For FN plots, there is a spreadsheet-based [18] 'orthodoxy test' that can filter out non-ideal datasets; a version for MG plots will be described elsewhere in due course.

A third major problem is the following. For ideal real emitters, even if one assumes the emitter radius is large enough for the SN barrier to be an adequate approximation for evaluating tunnelling probabilities, one expects that the extracted value $\{A_f^{SN}\}^{extr}$ would depend on emitter shape and on the applied-voltage range. There is already material in the literature that shows that this must be the case (e.g. [19–21]). However, the FN and MG plot theories are built using 'smooth planar emitter methodology'. In this approach, $A_f^{SN}$ is treated as if it were a constant, with the extracted value $\{A_f^{SN}\}^{extr}$ derived—with varying degrees of precision—from the slope and intercept of a straight line fitted to an experimental FN or MG plot.

My current understanding is that $\{A_f^{SN}\}^{extr}$ (as derived from an FN or MG plot) is actually some kind of effective average value of $[I_m/J_k^{SN}]$, taken over the range of $F_C$-values used in the experiments. But detailed physical interpretation of $\{A_f^{SN}\}^{extr}$ is an issue separate from whether the value extracted from an MG plot is a useful characterization parameter (which it is considered to be). Values of $\{A_f^{SN}\}^{extr}$ are presumed particularly useful for LAFEs, when comparing the properties of different emitting materials or processing regimes. Thus, having a simple method of extracting a numerically well-defined value (from a particular set of ideal experimental data) is expected to be helpful.

For LAFEs, a more useful property is perhaps the *extracted formal area efficiency* $\{\alpha_f^{SN}\}^{extr}$ (for the SN barrier), defined by

$$\{\alpha_f^{SN}\}^{extr} \equiv \frac{\{A_f^{SN}\}^{extr}}{A_M}, \tag{2.4}$$

where $A_M$ is the *LAFE macroscopic area* (or *footprint*). Few experimental values have been reported for $\{\alpha_f^{SN}\}^{extr}$. It is thought [22] to be very variable as between LAFEs, but perhaps to often lie in the vicinity of $10^{-7}$ to $10^{-4}$. Clearly, if—for some particular LAFE material—data analysis showed (for example) that apparently only $10^{-5}\%$ of the footprint area was actually emitting electrons, then this might indicate scope for practical improvements. This parameter looks potentially useful for technology development.

# 3. Theory of Murphy–Good plots

Given the above context, MG-plot theory can now be developed. This is most easily done using *scaled parameters and equations*, as follows. The *scaled (barrier) field $f$* (for a barrier of zero-field height $\phi$) is a dimensionless physical variable formally defined, using the *Schottky constant $c_S$* [$\equiv (e^3/4\pi\varepsilon_0)^{1/2}$] [10], by

$$f \equiv c_S^2 \phi^{-2} F_L \cong [1.439965 \text{ eV}^2 \text{ (V/nm)}^{-1}]\phi^{-2}F_L. \tag{3.1}$$

For a SN barrier of zero-field height $\phi$, the criterion $f = 1$ defines a *reference field* $F_R[= c_S^{-2}\phi^2]$ at which the barrier top is pulled down to the Fermi level. For this barrier, $f = F_L/F_R$, and hence $F_L = f F_R$. It can be shown from [8] (but, better, see arXiv:1801.08251v2) that $v_F = v(x = f_C)$, where $f_C = F_C/F_R$.

Scaling parameters $\eta(\phi)$ and $\theta(\phi)$ are defined by

$$\eta(\phi) \equiv bc_S^2\phi^{-1/2}, \quad \theta(\phi) \equiv ac_S^{-4}\phi^3. \tag{3.2}$$

Substituting $F_C = f_C F_R = f_C \, c_S^{-2}\phi^2$ into equation (2.2), and writing $v_F$ explicitly as $v(f_C)$, yields the *scaled equation*

$$I_m(f_C) = A_f^{SN}\theta(\phi)f_C^2 \exp\left[\frac{-\eta(\phi) \cdot v(f_C)}{f_C}\right]. \tag{3.3}$$

For simplicity, we now normally cease to show the dependence of $\eta$ and $\theta$ on $\phi$.

The parameter $f_C$ is helpful in characterizing FE theory and the behaviour of field emitters. For example, in the case of tungsten field emitters (with $\phi = 4.50$ eV) it is known that: (a) these emitters

most commonly operate within the $f_C$-range $0.15 < f_C < 0.35$ (see electronic supplementary material spreadsheet related to [18]), (b) the safe operating limit for pulsed emission (in traditional field electron microscope configuration) is about $f_C < 0.6$ [23,24], and (c) the derivation of the MG zero-temperature FE equation breaks down above about $f_C \approx 0.8$ [7]. Slightly different $f_C$-values would apply to materials with work-function different from 4.50 eV. Scaled-field values are easily converted back to local-field values by multiplying by the reference field $F_R$, which is approximately equal to 14.1 V nm$^{-1}$ for a $\phi = 4.50$ eV emitter.

A key development [25], in 2006, was the discovery of a simple good approximation for $v(f)$:

$$v_F = v(f) \approx 1 - f + (1/6)\, f \ln f. \tag{3.4}$$

In $0 \le f \le 1$, $v_F$ takes values between $v(f=0) = 1$ and $v(f=1) = 0$. For equation (3.4), in $0 \le f \le 1$, Forbes & Deane [8] found the maximum error in $v(f)$ as 0.0024 and the maximum percentage error as 0.33%. High-precision numerical formulae for $v(f)$, with maximum error $8 \times 10^{-10}$ in $0 \le f \le 1$, are also now known (see appendix B).

Setting $f = f_C$ and substituting equation (3.4) into equation (3.3) leads, after some rearrangement, to

$$I_m(f_C) \approx A_f^{SN} \cdot \theta \exp \eta \cdot f_C^\kappa \cdot \exp\left[\frac{-\eta}{f_C}\right], \tag{3.5a}$$

and

$$k \equiv 2 - \eta/6. \tag{3.5b}$$

For an ideal device/system, equation (3.1) can be used to define, by $V_{mR} = F_R \zeta_C$ $[= c_S^{-2} \phi^2 \zeta_C]$, a reference measured-voltage $V_{mR}$ at which, at location 'C', the SN barrier-top is pulled down to the Fermi level. It follows that

$$f_C = \frac{F_C}{F_R} = \frac{V_m/\zeta_C}{V_{mR}/\zeta_C} = \frac{V_m}{V_{mR}}, \tag{3.6}$$

and that equation (3.5a) can be rewritten as

$$I_m(V_m) \approx \{A_f^{SN} \cdot (\theta \exp \eta) \cdot V_{mR}^{-\kappa}\} \cdot V_m^\kappa \cdot \exp\left[\frac{-\eta V_{mR}}{V_m}\right], \tag{3.7}$$

and then

$$\ln\{I_m/V_m^\kappa\} \approx \ln\{A_f^{SN} \cdot (\theta \exp \eta) \cdot V_{mR}^{-\kappa}\} - \frac{\eta V_{mR}}{V_m}. \tag{3.8}$$

This is an equation for a *theoretical Murphy–Good plot*.

Since $A_f^{SN}$ is being treated as constant, and all parameters on the right-hand side (except $V_m$) are constants, equation (3.8) is predicted to be a *straight line* with slope $S_{MG}$ and intercept $\ln\{R_{MG}\}$ given by

$$R_{MG} = A_f^{SN} \cdot (\theta \exp \eta) \cdot V_{mR}^{-\kappa}, \tag{3.9}$$

and

$$S_{MG} = -\eta V_{mR} = -b\phi^{3/2} \zeta_C. \tag{3.10}$$

The subscript 'MG' indicates that these parameters 'belong to' a theoretical MG plot. It further follows that

$$R_{MG} \cdot (|S_{MG}|)^\kappa = A_f^{SN} \cdot \theta \cdot \exp \eta \cdot \eta^\kappa = A_f^{SN} \cdot \theta \eta^2 \cdot \exp \eta \cdot \eta^{-\eta/6}. \tag{3.11}$$

From equations above, $\theta \eta^2 = ab^2 \phi^2$ $[\cong (7.192492 \times 10^{-5} \text{ A nm}^{-2} \text{ eV}^{-2})\phi^2]$. Thus, if $S_{MG}$ and $\ln\{R_{MG}\}$ are identified with the slope $S_{MG}^{fit}$ and intercept $\ln\{R_{MG}^{fit}\}$ of a straight line fitted to an experimental MG plot, the extracted values of the VCL $\zeta_C$, the reference measured voltage $V_{mR}$, and the formal

**Table 1.** Typical values of quantities appearing in the 'extraction formulae' (3.12) to (3.14).

| $\phi$ (eV) | $b\phi^{3/2}$ (V nm$^{-1}$) | $\eta$ | $\exp\eta \cdot \eta^{-\eta/6}$ | $\Lambda_{\mathrm{MG}}(\phi)$ (nm$^2$ A$^{-1}$) |
|---|---|---|---|---|
| 2.50 | 27.00 | 6.2210 | 75.62 | 29.42 |
| 3.00 | 35.49 | 5.6790 | 56.55 | 27.32 |
| 3.50 | 44.73 | 5.2577 | 44.85 | 25.31 |
| 4.00 | 54.65 | 4.9181 | 37.06 | 23.45 |
| 4.50 | 65.21 | 4.6368 | 31.54 | 21.77 |
| 5.00 | 76.37 | 4.3989 | 27.46 | 20.25 |
| 5.50 | 88.11 | 4.1942 | 24.34 | 18.89 |

emission area $A_{\mathrm{f}}^{\mathrm{SN}}$ are

$$\zeta_{\mathrm{C}}^{\mathrm{extr}} = -\frac{S_{\mathrm{MG}}^{\mathrm{fit}}}{b\phi^{3/2}}, \tag{3.12}$$

$$\{V_{\mathrm{mR}}\}^{\mathrm{extr}} = -\frac{S_{\mathrm{MG}}^{\mathrm{fit}}}{\eta} \tag{3.13}$$

and

$$\{A_{\mathrm{f}}^{\mathrm{SN}}\}^{\mathrm{extr}} = \Lambda_{\mathrm{MG}} \cdot R_{\mathrm{MG}}^{\mathrm{fit}} \cdot (|S_{\mathrm{MG}}^{\mathrm{fit}}|)^{\kappa}, \tag{3.14}$$

where the *emission area extraction parameter* $\Lambda_{\mathrm{MG}}$ *(when using an MG plot)* is given by

$$\Lambda_{\mathrm{MG}}(\phi) \equiv \frac{1}{(ab^2\phi^2) \cdot \exp\eta \cdot \eta^{-\eta/6}}. \tag{3.15}$$

An extracted area-efficiency value can be obtained from equations (2.4) and (3.14), and an extracted value of macroscopic FEF $\gamma_{\mathrm{M}}$ from equation (3.12) and the relation

$$\gamma_{\mathrm{M}}^{\mathrm{extr}} = \frac{d_{\mathrm{M}}}{\zeta_{\mathrm{C}}^{\mathrm{extr}}}, \tag{3.16}$$

where $d_{\mathrm{M}}$ is the system distance used to define the FEF and related macroscopic field $F_{\mathrm{M}}$.

Since expression (3.15) depends only on $\phi$, a table of $\Lambda_{\mathrm{MG}}(\phi)$-values is easily prepared with a spreadsheet. Some illustrative values are shown in table 1. $\Lambda_{\mathrm{MG}}(\phi)$ is only weakly dependent on $\phi$, so uncertainty in the true $\phi$-value should cause little error in the extracted value of formal emission area.

The consistency of the above approach has been checked by simulations that use a modified version of an already existing special-purpose spreadsheet that calculates values for the FE SMFs, using the high-precision numerical expressions given in [8] and appendix B. These MG-plot related simulations have also been compared with simulations based on the equivalent theory (set out in appendix A) for interpreting an FN plot by using the extended MG equation. In both cases, the simulations have been carried out for the characteristic-scaled-field range $0.15 \leq f_{\mathrm{C}} \leq 0.35$, for selected values of local emitter work function in the range $2.50 \leq \phi/\mathrm{eV} \leq 5.50$. (Emitters with $\phi = 4.50$ eV are often operated within this scaled-field range.) An annotated copy of the spreadsheet as used in the simulations is provided as downloadable electronic supplementary material; details of the simulations are given in appendix B.

In general terms, the simulations confirm that the MG plots will normally yield very consistent results for extracted values of the reference measured voltage $V_{\mathrm{mR}}$, the characteristic voltage conversion length (VCL) $\zeta_{\mathrm{C}}$, and the formal emission area $A_{\mathrm{f}}^{\mathrm{SN}}$ for the SN barrier. In these simulations, the parameters $V_{\mathrm{mR}}$ and $\zeta_{\mathrm{C}}$ are extracted with a consistency of 0.1% or better, and $A_{\mathrm{f}}^{\mathrm{SN}}$ with a consistency of better than 1.8%. The corresponding figures for the FN plot are around 2% and around 52%, respectively. These results clearly demonstrate the superiority of the MG plot.

With the MG plot, there are small discrepancies between the input values for the various parameters and the 'typical extracted values', as assessed by the extracted values corresponding to the scaled-field value $f_{\mathrm{C}} = 0.25$. These discrepancies are around 0.3% for $V_{\mathrm{mR}}$ and $\zeta_{\mathrm{C}}$, and up to 1.8% for $A_{\mathrm{f}}^{\mathrm{SN}}$, and are thought to arise because MG plot theory is based on the simple good approximation (3.4), which is not an exact expression for the function $v(f)$.

For an FN plot, as interpreted via the EMG equation, the corresponding discrepancies between the input values and 'typical extracted values' are around 0.7% for $V_{\mathrm{mR}}$ and $\zeta_{\mathrm{C}}$, and up to 19% for $A_{\mathrm{f}}^{\mathrm{SN}}$.

So, again, the performance of the MG plot is significantly superior to that of the FN plot.

It needs to be understood that the numerics presented here have been generated specifically for the purpose of making numerical comparisons between the performances of MG plots and FN plots, and are considered to be 'validly indicative'. If different values had been used for the ranges of $f_C$-values and/or work functions employed in the simulations, then numbers slightly different from those reported above would have been generated. However, there is no reason to think that qualitative conclusions about the comparison of MG plots and FN plots would be affected.

It is also important that one should not take the numerics given here as good estimates of the likely errors involved when the extraction procedures discussed here are applied to real experimental results. Further factors come into play when real experimental results are involved, including noise in the experimental data, possible uncertainty in the true work-function value, and weaknesses in the 'smooth planar emitter' methodology that underlies both FN plots and MG plots. (Obviously, real emitters are very often shaped like rounded posts or pointed needles, and have atomic structure.) In the author's view, we currently have no adequate knowledge about the sizes of likely errors of this kind. The investigation of alternative $I_m(V_m)$ data interpretation methodologies and the likely errors involved are active topics of research (e.g. [19–21]).

# 4. Discussion

The essential merit of the MG plot is that the whole tiresome apparatus [11,12] of slope and intercept correction factors, fitting points and chord corrections (needed for high-precision parameter extraction when an FN plot is used with the EMG equation) has been swept away.

The author's view is that using MG plot analysis techniques based on the EMG equation should benefit three groups of experimentalists who currently use FN plots (and will also benefit the subject as a whole). Those who already use FN-plot interpretation theory based on equation (3.4) will no longer need to use slope and intercept correction factors, or equivalent. Those who already use the MG equation, but use formulae based ultimately on 1970s approximations for $v_F$, such as those of Spindt *et al.* [26] or Shrednik [27], will get slightly more precise results than before, and will not have to use approximation formulae whose true origin may not always be obvious.

However, the largest group of beneficiaries should be those who analyse FN plots by using the elementary FN-type equation (see [16]), which is a simplified version (see appendix A) of the original 1928 FN FE equation, with both equations based on assuming that the tunnelling barrier is ET. For this group, for *ideal* devices/systems, the simple formulae provided here allow them to precisely extract (from an MG plot) information about three characterization parameters (the VCL, the FEF and the formal area efficiency), rather than the current normal practice of extracting only one (the FEF).

In describing these extracted results using MG plots as 'precise', I refer primarily to the removal of the procedural and mathematical imprecisions associated with the use of FN plots and/or the use of 1970s era approximations for $v(f)$, $s_t$ and $r_t$. There remains, of course, the possibility of physical error due to incorrect choice of emitter work function when converting experimentally determined slope and intercept values to characterization parameters, using the extraction formulae (3.12)–(3.14). The sizes of the errors relating to particular pairs of correct and incorrect work-function values can be estimated roughly from table 1, which shows values for the quantities that appear in these extraction formulae, for selected work-function values. More precise estimates can be obtained by using the spreadsheet: inserting a work-function value into cell K19 will generate relevant quantity values in cells K25, K31 and K41.

One reviewer has suggested that it might be possible to overcome the above problem by applying multi-parameter numerical fitting to derive a work-function value. It is shown in the reviewer's report that this technique works effectively when applied to precisely simulated data, using the Matlab routine 'fminsearch'. This is an interesting suggestion that deserves to be explored further by additional simulations—but I fear that the technique may work less effectively when applied to noisy data such as may be collected in FE experiments—a point made to me by Kyritsakis (A. Kyritsakis 2019, private communication).

The following point also deserves note. Using either the original 1928 FN equation or the elementary FE equation to extract an area-like parameter from an FN plot would result in a formal-area estimate ($A_f^{ET}$) greater than $A_f^{SN}$ by a factor of typically around 100 (see appendix A). Taking the tunnelling barrier to be a SN barrier is 'better physics' [5] than taking it to be the ET barrier used in deriving the elementary FE equation. Hence one expects that extracting the area $A_f^{SN}$ should be 'better scientific procedure' than extracting the area $A_f^{ET}$.

The formulae here envisage that researchers will use their raw $I_m(V_m)$ data to make $I_m(V_m)$ MG plots, and will then apply an orthodoxy test [18]—which must be passed if values for extracted (and related) characterization parameters are to be regarded as trustworthy. As indicated earlier, an orthodoxy test already exists for FN plots, and a modified version [28] will be made available shortly for MG plots. Hopefully, this should help to reduce the incidence of spuriously high FEF values reported in the literature.

Using $I_m(V_m)$-type MG plots could also help eliminate the widespread but unfortunate literature practice of pre-converting $I_m(V_m)$ data to become $J_M(F_M^{app})$ data before making an FN plot, where $F_M^{app}$ is the apparent macroscopic field obtained from the pre-conversion equation, and $J_M$ is the *macroscopic (or LAFE-average) current density* defined by $J_M = I_m/A_M$. This pre-conversion is almost always carried out by using a plausible but often *defective* conversion equation (defective because it can be invalid for non-ideal devices/systems) [13]. This, in turn, has often led to defective FN plots and spurious results for characterization parameters.

Another feature of experimental FE literature is that papers sometimes use *macroscopic* current densities to show data or make FN plots, but state a formula for *local* current density in the text, without drawing attention to the difference. This practice creates un-discussed apparent discrepancies between theory and experiment, sometimes by a factor of $10^6$ or more. Such confusions would be reduced if, instead, FE papers gave an equation for measured current, either an $I_m(F_C)$ equation of form (2.2) above, or a related $I_m(V_m)$ equation.

The question also arises of how improved data-analysis theory of the general kind described in this paper might be applied to non-metals, in particular semiconductors and carbon-based materials such as carbon nanotubes (CNTs). For the last 90 years or so, it has been near-universal practice among FE experimentalists to apply 'smooth planar emitter methodology' and FN-plot theory to all materials 'as a first approximation', notwithstanding that this approach was originally developed to apply to a Sommerfeld free-electron metal. The introduction of MG plots does not change this situation: MG plots can be applied to all materials 'as a somewhat improved first approximation'.

The problem, of course, is how to do better than this. With FN plots, it is known (certainly to the author) that differences in surface exchange and correlation effects, as between metals and other materials, can in principle be represented by introducing new forms of slope and intercept correction factors, to replace $s_t$ and $r_t$. But this is rarely if ever done. The equivalent in the present work would be to introduce a different form of data plot in which $\kappa$ is taken to have a value intermediate between $(2 - \eta/6)$ and 2, but good relevant theory to decide this new value of $\kappa$ is not available in the literature, as far as I am aware.

A more serious difficulty, for both semiconductors and nanotubes, is the possibility of field penetration into the emitting material: this could make the operative work function $\phi^{op}$ depend significantly on the apex field $F_a$, and would require modification of the theory given here. At present, the possibility of doing this reliably is limited by the lack of good knowledge as to what the functional form of $\phi^{op}(F_a)$ would be for non-metal field emitters, in various circumstances.

It is also needful to remember that all FN and MG plots implicitly involve the (unrealistic) 'smooth planar emitter' methodology. As noted earlier, the issue of how best to include emitter-shape effects, when predicting FE $I_m(V_m)$ characteristics or analysing experimental FE $I_m(V_m)$ data, is a topic of active research (e.g. [19–21]). At present, no general agreement exists on how best to perform data analysis for non-planar emitters, and significant amounts of detailed further research seem needed.

Strategically, it seems more urgent to develop $I_m(V_m)$ data interpretation theory for point-form metal emitters than to examine how to apply 'smooth planar emitter' methodology to non-metals. Thus, for all the above reasons, detailed discussion of customized $I_m(V_m)$ data interpretation theory for non-metals seems premature, and is outside the scope of this paper.

Development of data interpretation theory for point-form emitters will inevitably require us to eventually move on from MG plots. An early step will be to examine more general data-plot forms that might be predicted to be linear or approximately linear, in particular the so-called power-$\kappa$ (or power-$k$) plot [21]. But, very probably, $I_m(V_m)$ data analysis will eventually find it useful or necessary to employ some more-sophisticated analysis technique, such as multi-parameter numerical fitting. This technique has been widely used outside the context of FE for many years, and sometimes within it. It potentially offers greater flexibility and greater precision in parameter extraction.

The author's view is that it is likely to be some years before $I_m(V_m)$ data-interpretation methodologies specifically designed for point-form emitters (including basic theory, easy-to-use validated tools, and any related knowledge needed to interpret or use their outputs) become widely available. In particular, it would ideally need to be shown that the methodologies work robustly for 'noisy' data inputs, can output 'measured' values of characteristic local field and scaled field, and can provide the equivalents of an orthodoxy test [18] and (desirably) 'phenomenological adjustment' [16] .

Until this happens, MG plots (which are straightforward to implement, and—like FN plots—are robust against moderate amounts of noise) can provide a significantly better approach to FE $I_m(V_m)$ data analysis than do FN plots.

Data accessibility. This article has no additional data.

Competing interests. I declare I have no competing interests.

Funding. We received no funding for this study.

Acknowledgements. Research by Dr Eugeni O. Popov and colleagues at the Ioffe Institute in Saint-Petersburg has been a major stimulus for this work. Their numerical simulations (e.g. [29]) relating to carbon nanotubes have looked for the value of $k$ in the empirical FE equation $I_m = CV_m^k \exp[-B/V_m]$ proposed [30] some years ago. My thinking about how to find $C$-values has led to this article. I thank Dr Popov for numerous e-mail discussions.

# Appendix A. Emission area extraction parameters for Fowler–Nordheim plots

This appendix gives expressions for emission area extraction parameters for FN plots. First consider the case where an FN plot is interpreted by (a) assuming that the tunnelling barrier is a SN barrier and (b) using the EMG equation. The current–voltage form of this equation is obtained by combining equations (2.2) and (2.3) above. In natural FN coordinates this becomes

$$\ln\{I_m^{EMG}/V_m^2\} = \ln\{A_f^{SN} a\phi^{-1}\zeta_C^{-2}\} - v_F b\phi^{3/2}\zeta_C/V_m). \tag{A 1}$$

Customizing the general theory in [11] yields the slope $S^{tan}$ of the tangent to the 'theoretical' plot (A 1) as

$$S^{tan}(V_m^{-1}) = -s(f_C) \cdot b\phi^{3/2}\zeta_C, \tag{A 2}$$

where $f_C$ [$=V_m/V_{mR}$] is the characteristic scaled-field value corresponding to measured voltage $V_m$, $V_{mR}$ is the reference measured voltage as discussed in the main text, and $s(f)$ is the slope correction function for the SN barrier, as usually defined (e.g. [8]). Also, from [11], the intercept $\ln\{R^{tan}\}$ that this tangent makes with the vertical ($1/V_m = 0$) axis is given via

$$R^{tan}(V_m^{-1}) = r(f_C) \cdot A_f^{SN} a\phi^{-1}\zeta_C^{-2}, \tag{A 3}$$

where $r(f_C)$ is the 2012 intercept correction function as defined in [11] and denoted there by $r_{2012}$.

Because a theoretical FN plot of the EMG equation is slightly curved, its slope (and hence the slope of its tangent) vary with the horizontal-axis coordinate $V_m^{-1}$. The *tangent method* of plot interpretation takes a given experimental FN plot to be parallel to this theoretical tangent as defined at a particular $V_m^{-1}$-value and hence at a particular $f_C$-value $f_t$. *Fitting values* of the correction functions are then defined by $s_t = s(f_t)$ and $r_t = r(f_t)$. On identifying the related values of $S^{tan}$ and $\ln\{R^{tan}\}$ with the slope $S_{FN}^{fit}$ and intercept $\ln\{R_{FN}^{fit}\}$ of the straight line fitted to the experimental FN plot, we find that

$$R_{FN}^{fit}|S_{FN}^{fit}|^2 = (r_t s_t^2)(ab^2\phi^2)A_f^{SN}. \tag{A 4}$$

Hence, the extracted value of $A_f^{SN}$ is given in terms of $S_{FN}^{fit}$ and $R_{FN}^{fit}$ by the extraction equation

$$\{A_f^{SN}\}^{extr} = \Lambda_{FN}^{SN} \cdot (R_{FN}^{fit}|S_{FN}^{fit}|^2), \tag{A 5}$$

where the *extraction parameter $\Lambda_{FN}^{SN}$ for an FN plot, interpreted by assuming a SN barrier*, is given by

$$\Lambda_{FN}^{SN} = \frac{1}{(ab^2\phi^2)(r_t s_t^2)}. \tag{A 6}$$

The fitting value $f_t$ is not initially known. In principle, it can be estimated by an iterative process, but normal practice takes $s_t = 0.95$ as a first approximation. This corresponds to $f_t \cong 0.2815$ and (for an emitter with work-function 4.500 eV) to $r_t s_t^2 \cong 112.9$. The corresponding extraction-parameter value is

$$\Lambda_{FN}^{SN} \sim 6.083\,\text{nm}^2\,\text{A}^{-1}. \tag{A 7}$$

If, instead, an FN plot is interpreted by assuming the tunnelling barrier is ET, then a numerically different result is found for the related extraction parameter $\Lambda_{FN}^{ET}$. In this case, an 'extended elementary

(EEL) equation' [16] is written in the current–voltage form

$$I^{EEL}(V_m) = A_f^{ET} a \phi^{-1} \zeta_C^{-2} V_m^2 \exp\left[\frac{-b\phi^{3/2}\zeta_C}{V_m}\right], \tag{A 8}$$

where $A_f^{ET}$ is the formal emission area for the ET barrier. The extracted value of $A_f^{ET}$ is given in terms of $S_{FN}^{fit}$ and $R_{FN}^{fit}$ by

$$\{A_f^{ET}\}^{extr} = \Lambda_{FN}^{ET} \cdot (R_{FN}^{fit}|S_{FN}^{fit}|^2), \tag{A 9}$$

where

$$\Lambda_{FN}^{ET} = \frac{1}{ab^2\phi^2}. \tag{A 10}$$

This result is found from equation (A 6) by noting that, for the ET barrier, $r_t$ and $s_t$ are both replaced by unity. For $\phi = 4.500$ eV, equation (A 10) yields

$$\Lambda_{FN}^{ET} \sim 686.6 \text{ nm}^2/\text{A}. \tag{A 11}$$

To achieve numerical consistency in making comparisons, values (A 7) and (A 11) are given here to four significant figures, but the physical precision is very much worse, particularly for value (A 7), which could easily be in error by 10% or more.

Clearly, for a given value of the experimentally derived product $(R_{FN}^{fit}|S_{FN}^{fit}|^2)$, use of the extraction-parameter value (A 11) will lead to estimates of the formal emission area $A_f^{ET}$ that are much larger (by a factor of order 100) than those found by using the extraction-parameter value (A 7) to estimate the formal emission area $A_f^{SN}$. Qualitatively, this is not surprising, since it is known (e.g. [1]) that the 1956 MG FE equation predicts emission current densities that are larger than those predicted by the elementary FE equation, by a factor typically between 250 and 500. This result underlines the need for careful definition of area-like quantities.

More important is the following conclusion. As shown in appendix B, extracted values of $A_f^{SN}$ found by analysing an MG plot are much the same as the extracted values of $A_f^{SN}$ found by using the extended MG equation to analyse an FN plot. This means that extracted values of $A_f^{SN}$ found by analysing an MG plot are much smaller (by a factor of order 100) than extracted values of $A_f^{ET}$ found by using the extended elementary equation to analyse an FN plot. Both these analysis procedures are relatively straightforward. However, when one accepts (for reasons discussed in [1]) that assuming a SN barrier is better physics than assuming an ET barrier, then the conclusion is that $A_f^{SN}$ is physically a 'more meaningful parameter' than $A_f^{ET}$, and that extracting an $A_f^{SN}$-value rather than an $A_f^{ET}$-value is 'significantly better scientific procedure'.

# Appendix B. Description and discussion of simulation procedures and results

This appendix describes simulations carried out in order to test the methodology proposed in this paper for extracting $A_f^{SN}$ values from a Murphy–Good plot, and to compare the precision of the methodology with that of the corresponding procedure for extracting $A_f^{SN}$ values from an FN plot. For simplicity, these simulations make use of an already existing special-purpose spreadsheet able to evaluate high-precision values of the FE SMFs $v(x)$ and $u(x)$ [$\equiv -dv/dx$] (and hence of all the FE SMFs, and of related quantities such as emission current densities). The parameter $x$ is the *Gauss variable* (i.e. the independent variable in the Gauss hypergeometric differential equation). These two functions are estimated by the following series, derived from those given in [8] by replacing the symbol $l'$ by the symbol $x$ now preferred, and by slightly adjusting the form of the resulting series for $v(x)$ (without changing its numerical predictions):

$$v(x) \cong (1-x)\left(1 + \sum_{i=1}^{4} p_i x^i\right) + x \ln x \sum_{i=1}^{4} q_i x^{i-1} \tag{B 1}$$

and

$$u(x) \cong u_1 - (1-x)\sum_{i=0}^{5} s_i x^i - \ln x \sum_{i=0}^{4} t_i x^i. \tag{B 2}$$

**Table 2.** Numerical constants for use in connection with equations (B 1) and (B 2).

| $i$ | $p_i$ | $q_i$ | $s_i$ | $t_i$ |
|---|---|---|---|---|
| 0 | — | — | 0.053 249 972 7 | 0.187 5 |
| 1 | 0.032 705 304 46 | 0.187 499 344 1 | 0.024 222 259 59 | 0.035 155 558 74 |
| 2 | 0.009 157 798 739 | 0.017 506 369 47 | 0.015 122 059 58 | 0.019 127 526 80 |
| 3 | 0.002 644 272 807 | 0.005 527 069 444 | 0.007 550 739 834 | 0.011 522 840 09 |
| 4 | 0.000 089 871 738 11 | 0.001 023 904 180 | 0.000 639 172 865 9 | 0.003 624 569 427 |
| 5 | — | — | −0.000 048 819 745 89 | — |

$u_1 = 3\pi/8\sqrt{2} \cong 0.8330405509$

Values of the constant coefficients $p_i$, $q_i$, $s_i$ and $t_i$ are shown in table 2.

It is readily seen that, at the values $x = 0,1$, equation (B 1) generates the exactly correct values $v(0) = 1$, $v(1) = 0$, and that at $x = 1$, equation (B 2) generates the exactly correct value $u(1) = u_1 = 3\pi/8\sqrt{2}$.

The form of equation (B 1) mimics the form of the lower-order terms in the (infinite) exact series expansion for $v(x)$ [6], but the coefficients in table 2 have been determined by numerical fitting to exact expressions for $v(x)$ and $u(x)$ (in term of complete elliptic integrals) evaluated by the computer algebra package MAPLE™. In the range $0 \leq x \leq 1$ (but not outside it), $v(x)$ takes values lying in the range $1 \geq v(x) \geq 0$, and the maximum error associated with formulae (B 1) and (B 2) is known to be less than $8 \times 10^{-10}$ [8]. The accuracy of the spreadsheet implementation is expected to be similar (e.g. see Wikipedia entry on 'Numeric precision in Microsoft Excel').

These formulae are applied in the context of Murphy–Good-type FE equations by setting $x = f_C$. A copy of the modified spreadsheet, as used in the present simulations, is provided as electronic supplementary material and will need to be downloaded.

For these simulations, the FE device/system has been taken as ideal, the local work function $\phi$ has been taken as 4.50 eV, the input value of the SN-barrier formal emission area $A_f^{SN}$ has been taken as constant and equal to 100 nm$^2$, and the input value of the reference measured voltage $V_{mR}$ has been taken as constant and equal to 6000 V. For a work-function value of 4.500 eV, this $V_{mR}$ value is equivalent to a constant characteristic voltage conversion length $\zeta_C$ of approximately 426.66 nm.

It is known (see spreadsheet in electronic supplementary material related to [18]) that tungsten field emitters (with assumed work function 4.50 eV) normally operate within the range $0.15 \leq f_C \leq 0.35$. In this range, for $f_C$-values increasing by steps of 0.01, values have been calculated (in the spreadsheet related to the present paper) for the measured voltage $V_m$ (column AM), its reciprocal $V_m^{-1}$ (column AQ), the characteristic kernel current density $J_{kC}^{SN}$ (column AN), the predicted measured current $I_m^{EMG}$ (column AO), and the MG-plot vertical-axis quantity $\ln\{I_m^{EMG}/V_m^\kappa\}$ (column AR).

For each of the $f_C$ values in the range $0.20 \leq f_C \leq 0.30$, an 'extracted local slope' $S_{MG}$ has been estimated (column AS) by using the equation

$$S_{MG} \approx \frac{Y(f_C - 0.05) - Y(f_C + 0.05)}{X(f_C - 0.05) - X(f_C + 0.05)}, \tag{B 3}$$

where $X[\equiv 1/V_m]$ and $Y[\equiv \ln\{I_m^{EMG}/V_m^\kappa\}]$ are the quantities on the horizontal and vertical axes of the MG plot. The parameter $S_{MG}$ given by equation (B 3) is the average slope over a scaled-field range of 0.1, centred on the chosen $f_C$-value. $S_{MG}$ is then used to derive an estimate (column AT) for the vertical-axis ($1/V_m = 0$) intercept $\ln\{R_{MG}\}$ of the tangent to the MG plot at the chosen $f_C$-value, using a formula equivalent to

$$\ln\{R_{MG}\} \approx Y(f_C) + |S_{MG}|X(f_C). \tag{B 4}$$

In order to make comparisons with extraction procedures that use an FN plot (as interpreted using the EMG equation) to estimate a value for $A_f^{SN}$, we have carried out manipulations similar to those just described, but with $\kappa$ taken equal to exactly 2. Columns BF and BG show the resulting values of $S_{FN}^{SN}$ and $\ln\{R_{FN}^{SN}\}$.

In relation to slope and intercept values extracted from the simulations, the observed near-constancy of the values in columns AS and AT shows that the MG plot is 'almost exactly' straight. The plot is not expected to be exactly straight, because MG plot theory is based on the 'simple good approximation' (3.4), which is not an exactly correct formula for $v(f_C)$.

Over the range of midpoint $f_C$-values considered, namely $0.20 \leq f_C \leq 0.30$, for the MG plot the variation in the extracted local slope is about 0.06% and that in the extracted intercept is about −0.1%. The corresponding figures for the FN plot are about 1.9% and about −2.0%. This confirms that the MG plot is much more closely linear than the FN plot.

For the parameters $\{V_{mR}\}^{extr}$ and $\zeta_C^{extr}$ that can be derived from the extracted slope, the derived variations are, of course, the same as the variations in the extracted slope. However, comparisons can also be made between the input value (6000 V for $V_{mR}$) and the extracted value $\{V_{mR}\}^{extr}$ for the central $f_C$-value in the whole range considered. For this value ($f_C = 0.25$), $\{V_{mR}\}^{extr}$ is 5982.5 V for the MG plot, 6041.6 V for the FN plot. These values quantify discrepancies between the input and extracted values of $V_{mR}$: for the MG plot the discrepancy is −0.29%, for the FN plot the discrepancy is +0.69%. For the MG plot, the discrepancy is probably caused by the use of the 'simple good approximation' to develop MG plot theory. The same figures and thinking apply to the extraction of characteristic VCL values, and to the extraction of characteristic FEF values via equation (3.16).

For the parameter $\{A_f^{SN}\}^{extr}$ extracted using an MG plot and equations (3.13.) and (3.14), the variation in this parameter over the midpoint range is about 1.3%, and the discrepancy between the input value and the central extracted value is about −1.4%. When this parameter is extracted using an FN plot and equations (A 5) and (A 6), the variation over the midpoint range is about 40% and the discrepancy between the input value and the central extracted value is about 15%. These figures confirm that, for the purpose of extracting a precise estimate of $A_f^{SN}$, the MG plot is demonstrably much superior to the FN plot.

The numerics presented here have been derived primarily for the purpose of comparing the merits of FN plots and MG plots. As noted in the main text, those for the MG plot should not be taken as good estimates of the likely errors involved in extracting characterization-parameter values from real experimental data.

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
