## [Reviewer comments · Royal Society Open Science]

Review History

RSOS-190912.R0 (Original submission)

Review form: Reviewer 1

Is the manuscript scientifically sound in its present form?

Yes

Are the interpretations and conclusions justified by the results?

Yes

Is the language acceptable?

Yes

Do you have any ethical concerns with this paper?

No

Have you any concerns about statistical analyses in this paper?

No

Recommendation?

Accept with minor revision (please list in comments)

Comments to the Author(s)

See attached (Appendix A).

Review form: Reviewer 2 (Stefania Carapezzi)**Is the manuscript scientifically sound in its present form?**

Yes

Are the interpretations and conclusions justified by the results?

Yes

Is the language acceptable?

Yes

Do you have any ethical concerns with this paper?

No

Have you any concerns about statistical analyses in this paper?

No

Recommendation?

Major revision is needed (please make suggestions in comments)

Comments to the Author(s)

Reviewer's comments to

"The Murphy-Good plot: a better method of analysing field emission data", R. Forbes
[Manuscript Number RSOS-190912]

The aim of the present paper is to provide the analytical tools to extract parameters from Murphy-Good (MG) plots of field emission (FE) current-voltage characteristics. The topic is presented in full scope, giving useful details over the calculations to be implemented, driving the reader across more than half of century of literature. Unfortunately, a major concern exists which would not recommend the acceptance of this work as it is.

Major comments:

The major point of the paper is to explain how to extract in reliable way parameters of interest, individuated in the characteristic voltage conversion length $\square C$ and the formal emission area AfsN, from FE current-voltage curves. To this aim it avails of MG plots (Eq. 3.8 of the manuscript), which are made linear by means of an approximation (Eq. 3.4 of the manuscript). By simulating a MG plot and then applying the illustrated procedure it is shown that the extracted value of AfsN lies within few percent from the actual value used in simulation. No mention has been made about the extracted $\square C$ value.

A first comment is that, in the reviewer's opinion, the procedure should have been tested not over the data produced by simulating a MG plot, that is Eq. 3.8, but over the simulated data got by the not approximated equation, that is Eq. 3.3, which in an ideal case would actually describe the current-voltage characteristics. If in fact this was the case, it should be stated in clearer terms in the manuscript, given that in it is mentioned that "The consistency of the above approach has been tested by simulating a MG plot".

The second comment is that in the extraction of both the αC and AfSN parameters enter the value of the work-function ϕ , which is rarely known exactly under the given experimental conditions. The influence of this uncertainty over the extracted values of αC and AfSN has not been assessed. The third point has been raised by the author himself in the last part of the Discussion Section, where he says that there could be the necessity in due time “to consider other analysis techniques, such as multi-parameter numerical fitting”. A Matlab code has been set up to perform a multi-parameter procedure to fit AfSN, αC and ϕ . By this code 1) data from Eq. 3.3 have computed and then 2) fitted by an optimization procedure performed through the Matlab routine `fminsearch`. Two model functions have been used to fit the simulated data: 2a) the one of Eq. 3.3, that is the MG equation without the approximation of Eq. 3.4 and 2b) the one of Eq. 3.8, the so called MG plot. In the following the results are shown: (please see the attached pdf file)

To interpret the figures above it should be considered that to generate the benchmark data it has been used the following values: AfSN = 100, αC = 0.2, ϕ = 4.5. The same (wild) guess of AfSN = 200, αC = 0.7, ϕ = 3 has been used as a starting point for the 1st run. Runs have been repeated until optimization has been reached, where 1st-guess parameters generated in the previous run have been used for all the runs after the 1st one. As it can be seen, it took the same number of runs to achieve the optimization. By using as model function Eq. 3.3 is possible to retrieve the exact values used to produce the benchmark data. Instead, by using as model function Eq. 3.8 (MG plot), the fitted values have been: AfSN = 97.9, αC = 0.22, ϕ = 4.17, that is the percentage of departure is of 2.1%, 12% and of 7%, respectively. It should be noticed that 1) by using a fitting procedure it is possible to extract all the three unknowns, AfSN, αC and ϕ , while by performing the procedure illustrated in the paper this is not possible, 2) even by using Eq. 3.8 with a fitting procedure some of the extracted values can present a departure up to 10%. Above all, there is no need to approximate Eq. 3.3, given there is no appreciable gain in computation time.

Minor typos:

In the abstract it is written (please see the attached pdf file (Appendix B))

For reference [14] is arXiv:1504.06134v7, and not arXiv:1504.01634v7.

Review form: Reviewer 3

Is the manuscript scientifically sound in its present form?

Yes

Are the interpretations and conclusions justified by the results?

Yes

Is the language acceptable?

Yes

Do you have any ethical concerns with this paper?

No

Have you any concerns about statistical analyses in this paper?

No

Recommendation?

Accept with minor revision (please list in comments)

Comments to the Author(s)

The author presents a method based on Murphy-Good plot to analyze field emission data. This method improves the conventional FN plot method. The author introduces the extracted formal

area efficiency (parameter) and the extracted values of VCL to connect the theoretical prediction and experimental data. They also modify the straight line from FN plot by R_MG and S_MG. This method is meaningful and valid for improving the experimental-data analysis. Therefore, I recommend to publish in R SOC.

It is better to add a discussion on the relation between the phenomenological parameters R_MG and S_MG and the properties of emitter, such as metallic or semiconducting or any new properties. This will provide more meaningful information for analyzing experimental field-emission data.

Decision letter (RSOS-190912.R0)

23-Jul-2019

Dear Dr Forbes,

The editors assigned to your paper ("The Murphy-Good plot: a better method of analysing field emission data") have now received comments from reviewers. We would like you to revise your paper in accordance with the referee and Associate Editor suggestions which can be found below (not including confidential reports to the Editor). Please note this decision does not guarantee eventual acceptance.

Please submit a copy of your revised paper before 15-Aug-2019. Please note that the revision deadline will expire at 00.00am on this date. If we do not hear from you within this time then it will be assumed that the paper has been withdrawn. In exceptional circumstances, extensions may be possible if agreed with the Editorial Office in advance. We do not allow multiple rounds of revision so we urge you to make every effort to fully address all of the comments at this stage. If deemed necessary by the Editors, your manuscript will be sent back to one or more of the original reviewers for assessment. If the original reviewers are not available, we may invite new reviewers.

- Data accessibility

If you wish to submit your supporting data or code to Dryad (<http://datadryad.org/>), or modify your current submission to dryad, please use the following link:
<http://datadryad.org/submit?journalID=RSOS&manu=RSOS-190912>

- Competing interests

- Authors' contributions

- Acknowledgements

- Funding statement

Kind regards,

Andrew Dunn

on behalf of Dr Chong Li (Associate Editor) and Miles Padgett (Subject Editor)
 openscience@royalsociety.org

Associate Editor's comments (Dr Chong Li):

Associate Editor: 1

Comments to the Author:

Please address all points raised by the reviewers especially the 3rd comment by Reviewer 2.

Comments to Author:

Reviewers' Comments to Author:

Reviewer: 1

Comments to the Author(s)

See attached

Reviewer: 2

Comments to the Author(s)

Reviewer's comments to

"The Murphy-Good plot: a better method of analysing field emission data", R. Forbes
 [Manuscript Number RSOS-190912]

The aim of the present paper is to provide the analytical tools to extract parameters from Murphy-Good (MG) plots of field emission (FE) current-voltage characteristics. The topic is presented in full scope, giving useful details over the calculations to be implemented, driving the reader across more than half of century of literature. Unfortunately, a major concern exists which would not recommend the acceptance of this work as it is.

Major comments:

The major point of the paper is to explain how to extract in reliable way parameters of interest, individuated in the characteristic voltage conversion length $\square C$ and the formal emission area AfsN, from FE current-voltage curves. To this aim it avails of MG plots (Eq. 3.8 of the manuscript), which are made linear by means of an approximation (Eq. 3.4 of the manuscript). By simulating a MG plot and then applying the illustrated procedure it is shown that the extracted value of AfsN lies within few percent from the actual value used in simulation. No mention has been made about the extracted $\square C$ value.

A first comment is that, in the reviewer's opinion, the procedure should have been tested not over the data produced by simulating a MG plot, that is Eq. 3.8, but over the simulated data got by the not approximated equation, that is Eq. 3.3, which in an ideal case would actually describe the current-voltage characteristics. If in fact this was the case, it should be stated in clearer terms in the manuscript, given that in it is mentioned that "The consistency of the above approach has been tested by simulating a MG plot".

The second comment is that in the extraction of both the $\square C$ and AfsN parameters enter the value of the work-function \square , which is rarely known exactly under the given experimental conditions. The influence of this uncertainty over the extracted values of $\square C$ and AfsN has not been assessed. The third point has been raised by the author himself in the last part of the Discussion Section, where he says that there could be the necessity in due time "to consider other analysis techniques, such as multi-parameter numerical fitting". A Matlab code has been set up to perform a multi-parameter procedure to fit AfsN, $\square C$ and \square . By this code 1) data from Eq. 3.3 have computed and then 2) fitted by an optimization procedure performed through the Matlab routine

fminsearch. Two model functions have been used to fit the simulated data: 2a) the one of Eq. 3.3, that is the MG equation without the approximation of Eq. 3.4 and 2b) the one of Eq. 3.8, the so called MG plot. In the following the results are shown: (please see the attached pdf file)

To interpret the figures above it should be considered that to generate the benchmark data it has been used the following values: $AfSN = 100$, $\square C = 0.2$, $\square = 4.5$. The same (wild) guess of $AfSN = 200$, $\square C = 0.7$, $\square = 3$ has been used as a starting point for the 1st run. Runs have been repeated until optimization has been reached, where 1st-guess parameters generated in the previous run have been used for all the runs after the 1st one. As it can be seen, it took the same number of runs to achieve the optimization. By using as model function Eq. 3.3 is possible to retrieve the exact values used to produce the benchmark data. Instead, by using as model function Eq. 3.8 (MG plot), the fitted values have been: $AfSN = 97.9$, $\square C = 0.22$, $\square = 4.17$, that is the percentage of departure is of 2.1%, 12% and of 7%, respectively. It should be noticed that 1) by using a fitting procedure it is possible to extract all the three unknowns, $AfSN$, $\square C$ and \square , while by performing the procedure illustrated in the paper this is not possible, 2) even by using Eq. 3.8 with a fitting procedure some of the extracted values can present a departure up to 10%. Above all, there is no need to approximate Eq. 3.3, given there is no appreciable gain in computation time.

Minor typos:

In the abstract it is written (please see the attached pdf file)

For reference [14] is arXiv:1504.06134v7, and not arXiv:1504.01634v7.

Reviewer: 3

Comments to the Author(s)

The author presents a method based on Murphy-Good plot to analyze field emission data. This method improves the conventional FN plot method. The author introduces the extracted formal area efficiency (parameter) and the extracted values of VCL to connect the theoretical prediction and experimental data. They also modify the straight line from FN plot by R_MG and S_MG . This method is meaningful and valid for improving the experimental-data analysis. Therefore, I recommend to publish in R SOC.

It is better to add a discussion on the relation between the phenomenological parameters R_MG and S_MG and the properties of emitter, such as metallic or semiconducting or any new properties. This will provide more meaningful information for analyzing experimental field-emission data.

Author's Response to Decision Letter for (RSOS-190912.R0)

See Appendix C.

RSOS-190912.R1 (Revision)

Review form: Reviewer 1

Is the manuscript scientifically sound in its present form?

Yes

Are the interpretations and conclusions justified by the results?

Yes

Is the language acceptable?

Yes

Do you have any ethical concerns with this paper?

No

Have you any concerns about statistical analyses in this paper?

No

Recommendation?

Accept as is

Comments to the Author(s)

The author has made a good faith effort to respond to all of the recommendations by this reviewer (and, in my judgment, the other reviewers as well). He has justified and defended his decisions appropriately and with good reasoning and arguments, as well as has made well-considered modifications to the manuscript itself. The new appendix is long, but I do not object to its inclusion, as there is a community it will serve. I am comfortable recommending the manuscript for publication.

Review form: Reviewer 2 (Stefania Carapezzi)

Is the manuscript scientifically sound in its present form?

Yes

Are the interpretations and conclusions justified by the results?

Yes

Is the language acceptable?

Yes

Do you have any ethical concerns with this paper?

No

Have you any concerns about statistical analyses in this paper?

No

Recommendation?

Accept as is

Comments to the Author(s)

The corrections made to the manuscript have met all the concerns of the reviewer. The suggestion is to accept the manuscript as is.

Decision letter (RSOS-190912.R1)

04-Nov-2019

Dear Dr Forbes,

I am pleased to inform you that your manuscript entitled "The Murphy-Good plot: a better method of analysing field emission data" is now accepted for publication in Royal Society Open Science.

on behalf of Dr Chong Li (Associate Editor) and Miles Padgett (Subject Editor)
openscience@royalsociety.org

Reviewer comments to Author:
Reviewer: 2

Comments to the Author(s)
The corrections made to the manuscript have met all the concerns of the reviewer. The suggestion is to accept the manuscript as is.

Reviewer: 1

Comments to the Author(s)
The author has made a good faith effort to respond to all of the recommendations by this reviewer (and, in my judgment, the other reviewers as well). He has justified and defended his decisions appropriately and with good reasoning and arguments, as well as has made well-considered modifications to the manuscript itself. The new appendix is long, but I do not object

to its inclusion, as there is a community it will serve. I am comfortable recommending the manuscript for publication.

Appendix A

Manuscript ID RSOS-190912

Title: "The Murphy-Good plot: a better method of analysing field emission data" Author: Forbes, Richard

Characterizing field emission data with respect to emitter work function, emission characteristics, and the extraction parameters that judge quality or that allow for predictive estimates of performance in devices (characterization setups may operate at lower or less demanding levels than how the emitters are used) or in understanding the nature of *undesired* field emission current as occurs in breakdown, is standard practice. And yet, as the author maintains and is rightly distressed by, it is often practice fraught with error at worst and sloppiness at best. The abstract gives both the spirit and intent of the manuscript well, and portends the advocacy slant therein. The manuscript is heavy with aesthetic appraisals and recommendations as to proper procedure. The author is passionate about correcting standard methods, and targets (as identified in the third paragraph of the conclusion) a particular set of "beneficiaries"

...who analyse FN plots by using the elementary FN-type equation... For this group, for ideal devices/systems, the simple formulae provided here allow them to precisely extract (from an MG plot) information about three characterization parameters, rather than one: the VCL, the FEF and the formal area efficiency.

Additionally, the author very clearly articulates (page 9, lines 25-33) a good and simple working model, for whose usage it is directed, and why it is important. Insofar as this is a reasonably large audience dominated by experimental practitioners, the manuscript serves a purpose to the community. The author has a particular viewpoint, and is relying primarily on the methods he personally has developed over two decades to provide a useful tool to the experimental community. Indeed, he is likely the strongest advocate of stringent methodology for how such methods are used. Parenthetically, this reviewer advises that the author's breadth of experience and wide-ranging knowledge make him uniquely qualified, and his self-evident affinity for the experimentalists make him well suited, for such a task. The manuscript more than meets the stated aims. The recommendation is therefore for publication.

Several comments and recommendations are made for the author's consideration.

1. Page 3, lines 27-36: the word *perverse* signals a passionately held aesthetic judgment (that one method is proper and others are deviant) but it is not shared. The author may prefer a notation motivated by a differential equation for the SN functions, but the development of Fowler-Nordheim-like equations and the WKB factors on which they depend does not demand it: other methods (unmentioned) with comparable accuracy do exist that have separate advantages and which use y to define an "effective" work function without reference to the SN functions. Acquiescence to the imputation condones it, and hides that usage is a matter of taste, not procedure. Personal advocacy can be indulged, but physics has many methods and toleration is pro forma. Moreover, the argument (without the pejorative) appears in the author's prior work with Deane.
2. Page 3, top: the author states

*his causes theoretical FN plots predicted by the MG equation to be slightly curved, rather than straight. This in turn causes very significant problems of detail [10], and the need for related procedures [11] when attempts are made to give well-defined precise meanings to the slope and intercept of the **straight line fitted** to a FN plot*

of experimental data. This article shows how to eliminate these problems, by finding a plot form that the MG equation predicts to be “almost exactly” linear. (emphasis added)

The author’s procedure accounts for curvature induced by $v(f)$ not being linear in f (Eq. 3.4) and how that ratchets through the behavior of J_L^{MG} , admittedly an important effect, but *not the only one* as the author concedes later. The author knows that geometry and statistics (either multiple emission locations and/or work functions on a single emitter, or multiple emitters with varying characteristics as in arrays or LAFE) introduce field-dependent quantities throughout any equation for $I(V)$ that are not accounted for by attention to a single cause of curvature. *All* experimental data are affected by multiple sources of non-linearity. The procedure to eliminate $v(f)$ effects is valuable; the prediction that it is sufficient to handle *all* experimental data effects is overstated.

3. Page 3, lines 25-29: the author speaks of an “uncertainty factor” λ that is unknown but argued to be constant, and makes reference to “current thinking” about its range of values. Elsewhere the author as enumerated what λ depends upon, but alternate methods constrain how such a coefficient behaves and give good accounts of experimental data, although the author’s audience would likely not avail themselves of them. The point, though, is that these other methods exist, and grouping them into an uncertainty factor *overestimates* the uncertainty and obscures that other approaches do not treat it as constant. Further, saying “current thinking” leads to an unwarranted inference of generality that the author has avoided elsewhere by properly stating “My” and “I”, such as when the author states “*I now prefer...*” (page 5, line 11) - which is perfect.
4. Page 4, line 60. The assumption that A_f is constant is part of the author’s procedure and argued later to be “helpful,” but as he acknowledges (page 5, line 47),

one expects that $A_f SN$ would depend on emitter shape and applied voltage. However, the FN and MG plot theories are built using the planar emission approximation. In this approach $A_f SN$ is treated as a constant, with the extracted value $A_f SN_{extr}$ derived – with varying degrees of precision – from the slope and intercept of a straight line fitted to an experimental FN or MG plot.

This is a particular way of proceeding that likely appeals to experimentalists, but with the non-specific “*one expects*”, the many approaches by numerous practitioners (albeit generally on the theory side) in the literature that do *not* make that approximation and consider the full dependence of field dependent area factors - and the significant advantages of doing so - is hidden, in the opinion of this reviewer, disadvantageously so. Constant area factors are a conscious but not unavoidable choice to a specific end. Alternate practice is more than expected, it is *done*.

5. Page 6, line 34, the author notes *the criterion $f = 1$ defines a reference field... at which the barrier top is pulled down to the Fermi level.* and returns several times below to the $f \approx 1$ condition. The FN equation, and even the MG variant of it, is *not* valid near that limit, it is a limit only of interest to the $v(f)$ SN function. For experimentalists, the caution that non-linearity due to neglected corrections to the transmission probability will matter and should not be wrongly attributed to something else. This is not just a caution to experimentalists: the theory literature likewise has its uncomfortable instances of when researchers have wrongly

posited fields that pull barriers below the Fermi level. The author typically notes bounds of applicability, but this instance could use a missing note.

6. The author makes several projections, *e.g.*, *Issues of how formal area relates to the notional area... are matters for separate discussion later, maybe in many years' time...* and *At present, no consensus exists on how best to perform data analysis for non-planar emitters, and significant amounts of detailed further research seem needed. It may take several years or more to reach consensus, and many further years to develop fully correct theory. Expectation (sic) is that, in due course (some, or likely many, years away), we shall need to move on from MG plots....* These are surprising statements for several reasons. First, it clearly lays out what is lacking in slope-intercept methods for analyzing data rather succinctly. Second, it acknowledges why more intensive methods that integrate local current density relations over specified surfaces (what the middle of Eq. (2.1) refers to) are needed and why they resolve such defects. Third, by saying “consensus” it implies methods presently being used are somehow suspect (the author means consensus as “preferred” which is correct, but it can also be taken as “agreed to be valid” which is misleading). But fourth, it implies that such efforts are in the future, possibly by years, even as the first three points make a strong case for why they should have been pursued all along. I argue that they are not far off, that they have existed for some time (since the late 1990’s) even if not in the form of easy computational tools favored or accepted by experimentalists, and (most importantly) that with modern computational power, reliance on slope-intercept methods to analyze $I(V)$ data is a choice, not a requirement. $I(V)$ curves can be readily generated (and routinely are) numerically and in some cases analytically by equations based on the middle of Eq. (2.1) resembling $\int J dA$. Coupled with modern multi-parameter fitting algorithms and methods that presently exist (such as the Statistical and Mathematical add-on packages in MATLAB or equivalent search methods in MATHEMATICA), high quality parameter extraction methods can be *presently employed* for those willing to invest the effort (a qualification that matters to the target audience). For some approaches, further development might be needed, but not for all, and in the later, the effort is not necessarily daunting. I would urge reconsideration of speculating on when advancements occur; I would favor an indication of what is possible with present methods, removing the imputation by the misleading statement (page 9, line 54) “*may possibly need to consider other analysis techniques, such as multi-parameter numerical fitting, rather than new plot forms.*” that they do not as yet exist.
7. In the conclusion, the author asserts “*The essential merit of the Murphy-Good plot is that the whole tiresome apparatus [10,11] of slope and intercept correction factors, fitting points and chord corrections (needed for **high precision** when a FN plot is used with the EMG or MG equations) has been **swept away...***” (emphasis added) Although the sentiment has merit, it is mild overstatement if it demands using “uncertainty factors” or relies on far off resolutions to conundrums. Other (more intensive) methods exist, and although they demand more than Excel, they deliver high precision, and I would argue they are far from replaceable.
8. Page 8, line 58: the author says “*...it is known... that error by a large factor – typically around 100 – is involved when this approach is used*” begs for an answer to “how?”. I presume the author means primarily by the inclusion of a factor of the form of e^η : for a work function of 4.5 eV, $\eta \approx 4.63$, or $e^\eta \approx 102.5$, something perhaps worth being explicit about.
9. The bibliography focuses almost exclusively on the author’s program for proper procedure

aimed at a particular group of experimentalists. That is not objectionable given the genesis of the methods. However, only acknowledging (page 9, line 49) “*The issues of how best to include emitter shape... are topics of active research, impracticable to summarize here.*” too easily fails to deliver on what those alternatives are or which developments are advantageous: the author’s (optional) views are germane. Either identify them or avoid mention as it is unnecessarily tantalizing.

Appendix B

Reviewer's comments to

"The Murphy-Good plot: a better method of analysing field emission data", R. Forbes
[Manuscript Number RSOS-190912]

The aim of the present paper is to provide the analytical tools to extract parameters from Murphy-Good (MG) plots of field emission (FE) current-voltage characteristics. The topic is presented in full scope, giving useful details over the calculations to be implemented, driving the reader across more than half of century of literature. Unfortunately, a major concern exists which would not recommend the acceptance of this work as it is.

Major comments:

The major point of the paper is to explain how to extract in reliable way parameters of interest, individuated in the characteristic voltage conversion length ζC and the formal emission area A_f^{SN} , from FE current-voltage curves. To this aim it avails of MG plots (Eq. 3.8 of the manuscript), which are made linear by means of an approximation (Eq. 3.4 of the manuscript). By simulating a MG plot and then applying the illustrated procedure it is shown that the extracted value of A_f^{SN} lies within few percent from the actual value used in simulation. No mention has been made about the extracted ζC value.

A first comment is that, in the reviewer's opinion, the procedure should have been tested not over the data produced by simulating a MG plot, that is Eq. 3.8, but over the simulated data got by the not approximated equation, that is Eq. 3.3, which in an ideal case would actually describe the current-voltage characteristics. If in fact this was the case, it should be stated in clearer terms in the manuscript, given that in it is mentioned that "The consistency of the above approach has been tested by simulating a MG plot".

The second comment is that in the extraction of both the ζC and A_f^{SN} parameters enter the value of the work-function ϕ , which is rarely known exactly under the given experimental conditions. The influence of this uncertainty over the extracted values of ζC and A_f^{SN} has not been assessed.

The third point has been raised by the author himself in the last part of the Discussion Section, where he says that there could be the necessity in due time "to consider other analysis techniques, such as multi-parameter numerical fitting". A Matlab code has been set up to perform a multi-parameter procedure to fit A_f^{SN} , ζC and ϕ . By this code 1) data from Eq. 3.3 have computed and then 2) fitted by an optimization procedure performed through the Matlab routine `fminsearch`. Two model functions have been used to fit the simulated data: 2a) the one of Eq. 3.3, that is the MG equation without the approximation of Eq. 3.4 and 2b) the one of Eq. 3.8, the so called MG plot. In the following the results are shown:

To interpret the figures above it should be considered that to generate the benchmark data it has been used the following values: $A_f^{SN} = 100$, $\zeta C = 0.2$, $\phi = 4.5$. The same (wild) guess of $A_f^{SN} = 200$, $\zeta C = 0.7$, $\phi = 3$ has been used as a starting point for the 1st run. Runs have been repeated until optimization has been reached, where 1st-guess parameters generated in the previous run have been used for all the runs after the 1st one. As it can be seen, it took the same number of runs to achieve the optimization. By using as model function Eq. 3.3 is possible to retrieve the exact values used to produce the benchmark data. Instead, by using as model function Eq. 3.8 (MG plot), the fitted values have been: $A_f^{SN} = 97.9$, $\zeta C = 0.22$, $\phi = 4.17$, that is the percentage of departure is of 2.1%, 12% and of 7%, respectively. It should be noticed that 1) by using a fitting procedure it is possible to extract all the three unknowns, A_f^{SN} , ζC and ϕ , while by performing the procedure illustrated in the paper this is not possible, 2) even by using Eq. 3.8 with a fitting procedure some of the extracted values can present a departure up to 10%. Above all, there is no need to approximate Eq. 3.3, given there is no appreciable gain in computation time.

Minor typos:

In the abstract it is written $\ln\{I_m/V_m^{(2-\eta/6)}\}$, instead of $\ln\{I_m/V_m^{(2-\eta/6)}\}$.

For reference [14] is arXiv:1504.06134v7, and not arXiv:1504.01634v7.

Appendix C

The Murphy-Good plot: a better method of analysing field emission data

Richard G. Forbes

*University of Surrey, Advanced Technology Institute & Dept. of Electrical and Electronic Engineering,
Guildford, Surrey, GU2 7XH, UK.*

e-mail: r.forbes@trinity.cantab.net

Keywords: Field electron emission (FE), current-voltage data interpretation, field emitter characterization, Murphy-Good theory, Fowler-Nordheim (FN) plots, Murphy-Good (MG) plots

Measured field electron emission (FE) current-voltage $I_m(V_m)$ data are traditionally analysed via Fowler-Nordheim (FN) plots, as $\ln\{I_m/V_m^2\}$ vs $1/V_m$. These have been used since 1929, because in 1928 FN predicted they would be linear. In the 1950s, a mistake in FN's thinking was found. Corrected theory by Murphy and Good (MG) made theoretical FN plots slightly curved. This causes difficulties when attempting to extract precise values of emission characterization parameters from straight lines fitted to experimental FN plots. Improved mathematical understanding, from 2006 onwards, has now enabled a new FE data-plot form, the "Murphy-Good plot". This plot has the form $\ln\{I_m/V_m^{(2-\eta/6)}\}$ vs $1/V_m$, where $\eta \cong 9.836239 \text{ (eV}/\phi)^{1/2}$ and ϕ is the local work function. Modern ("21st century") MG theory predicts that a theoretical MG plot should be "almost exactly" straight. This makes precise extraction of well-defined characterization parameters from ideal $I_m(V_m)$ data much easier. This article gives the theory needed to extract characterization parameters from MG plots, setting it within the framework of wider difficulties in interpreting FE $I_m(V_m)$ data (among them, use of "smooth planar emitter methodology"). Careful use of MG plots could also help remedy other problems in FE technological literature. It is suggested that MG plots should replace FN plots.

1. Background

Field electron emission (FE) occurs in many technological contexts, especially electron sources and electrical breakdown. A need exists for effective analysis of measured FE current-voltage [$I_m(V_m)$] data, to extract emission characterization parameters. These include: parameters that connect field to voltage; the field enhancement factors (FEFs) often used to characterize large-area field-electron emitters (LAFEs); and parameters relating to emission area and area efficiency (the latter being a measure of what fraction of emitter area is emitting significantly). This article proposes a **simple** new method for FE $I_m(V_m)$ data analysis, and urges its widespread adoption. This method, the Murphy-Good (MG) plot, **is in principle more precise than the** traditional Fowler-Nordheim (FN) plot. **The article represents one individual small part of a much wider project, partly outlined elsewhere [1], that aims to put FE theory onto a better scientific basis, by (amongst other things) improving the precision of interpretation of FE experimental data.**

To develop MG-plot theory efficiently, some **preliminary** discussion and refinement of traditional FN theory is needed. FN plots were introduced by Stern et al. [2] in 1929. They have the form $\ln\{I_m/V_m^2\}$ vs $1/V_m$ (or equivalent using other physical variables), and are used because the original 1928 FN equation [2] implied that FN plots of experimental data should be straight lines, with characterization data derivable from the slope and intercept.

However, in 1953, Burgess, Kroemer & Houston (BKH) [3] found a mathematical mistake in 1928 theoretical work by Nordheim [4], and a related physical mistake in FN's thinking. FN had assumed [2] that image-force rounding could be disregarded, and treated the electron tunnelling barrier as exactly triangular. BKH showed that rounding was much more important than FN had thought and Nordheim had calculated, and that (for emitters modelled as planar) it is *necessary* to base analyses on planar image-rounded barriers [often now called "Schottky-Nordheim" (SN) barriers]. Corrected analysis inserted a "barrier form correction factor" into the exponent of the original 1928 FN equation, and led to much higher tunnelling probabilities (**typically by a factor between 250 and 500** [5]). This correction factor is **generated** by an appropriate value of a special mathematical function (SMF) $v(x)$ now known [6] to be a special solution of the Gauss Hypergeometric Differential Equation. The *Gauss variable* x is the independent variable in this equation.

The Nordheim parameter y **used** in older FE discussions is given by $y = +x^{1/2}$, but its **use in mathematical contexts** can now be recognized as **illogical**—when a function " \mathcal{F} " is the solution of a differential equation, mathematics does not normally represent \mathcal{F} as a function of the square root of the independent variable in the equation. The use of y (rather than $x [=y^2]$) in FE literature is due to an unfortunate arbitrary choice (separate from the above mistake) made by Nordheim in his 1928 paper. Although y is useful as a modelling parameter in some theoretical discussions, hindsight indicates that

Richard Forbes 3/10/2019 13:18

Comment [1]: Reviewer 1, Point 1.2 (part)

Richard Forbes 3/10/2019 13:18

Comment [2]: Reviewer1, Point 1.1

Richard Forbes 7/9/2019 05:51

Deleted: *perverse*

choosing to use $x [=y^2]$ in 1928 would have proved better mathematics (and better for discussing FE $I_m(V_m)$ data).

In 1956, Murphy and Good (MG) [7] used the BKH results to develop a revised FE equation. [See Ref. [8] for a treatment that uses the modern "International System of Quantities" (ISQ) [9].] The zero-temperature version of their equation is called here the *Murphy-Good (MG) FE equation*. This equation is an adequate approximation at room temperature.

The MG FE equation gives the local emission current density (LECD) J_L^{MG} in terms of the local work function ϕ and local barrier field F_L . It is clearest to start from the linked form

$$J_L^{MG} = t_F^{-2} J_{kl}^{SN}, \quad (1.1a)$$

$$J_{kl}^{SN} = a\phi^{-1}F^2 \exp[-v_F b \phi^{3/2}/F_L], \quad (1.1b)$$

where $a [= 1.541434 \mu A eV V^{-2}]$ and $b [= 6.830890 eV^{-3/2} V nm^{-1}]$ are universal constants [10], often called the *first* and *second Fowler-Nordheim constants*, v_F is the value of $v(x)$ that applies to the SN barrier defined by ϕ and F_L , and t_F is the corresponding value of a special mathematical function $t(x)$ defined by

$$t(x) = v(x) - (4/3)x dv/dx. \quad (1.2)$$

J_{kl}^{SN} is called the *kernel current density for the SN barrier*, and can be evaluated *precisely* when ϕ and F_L are known.

The correction factor v_F is field-dependent (see below). This causes theoretical FN plots predicted by the MG FE equation to be slightly curved, rather than straight. This in turn causes very significant problems of detail and the need for related procedures, when attempts are made to give well-defined precise meanings to the slope and intercept of the straight line fitted to a FN plot of experimental data. These interpretation procedures involve correct choice of fitting point [11, 12] and application of a chord correction [12]. This article shows how to eliminate these particular problems, by finding a plot form that the MG FE equation predicts to be "almost exactly" linear.

An ideal FE device/system is one in which the measured current-voltage $I_m(V_m)$ characteristics are determined only by unchanging system geometry and by the emission process (see [13], and below). If curvature in an FN plot taken from an ideal FE device/system is due to physical reasons (such as small apex radius of curvature, or—with a LA FE—statistical variations in the characteristics of individual emitters), then use of a MG plot will not be able to straighten out this kind of curvature, though it should be a useful step forwards.

Richard Forbes 3/10/2019 13:19
Comment [3]: Reviewer 1, Point 1.2.

2. Some general issues affecting field emission current-voltage data analysis

In fact, three other major problems affect both FN-plot and MG-plot interpretation, and need discussion. The FN and MG derivations disregard the existence of atoms and model the emitter surface as smooth, planar and structureless. This *smooth planar emitter methodology* is unrealistic, but creating reliably better theory is very difficult, although there are some atomic-level treatments, e.g., Refs [14, 15]. When applying a current-density equation to real emitters, this weakness can be explicitly formalized, as follows.

To recognize both this difficulty and all other factors omitted in deriving eq. (1.1b), the present author has replaced t_F^{-2} in eq. (1.1a) by an "uncertainty factor" λ of unknown functional behaviour (see [16] for recent discussion). I now write $J_L^{\text{EMG}} = \lambda J_{\text{kl}}^{\text{SN}}$, and call the revised equation (and variants using other physical variables) the *Extended Murphy-Good (EMG) FE equation*. My current thinking [17] is that (for a SN barrier) λ varies with relevant parameters (in particular, local field) and most probably lies somewhere in the range $0.005 < \lambda < 14$. (though it could turn out, when further atomic-level treatments are available, that the figure 0.005 has been pessimistic and unnecessarily low [17]).

In principle, the related total emission current (I_e^{EMG}) is found by integrating J_L^{EMG} over the emitter surface and writing the result as first shown below, where A_n^{EMG} is the *notional emission area* (as derived using the EMG equation):

$$I_e^{\text{EMG}}(F_C) = \int J_L^{\text{EMG}} dA = A_n^{\text{EMG}} J_C^{\text{EMG}} = A_n^{\text{EMG}} \lambda J_{\text{kc}}^{\text{SN}} = A_f^{\text{SN}} J_{\text{kc}}^{\text{SN}}. \quad (2.1)$$

The subscript "C" denotes characteristic values taken at some characteristic location on the emitter surface (in modelling, nearly always the emitter apex).

The second form follows from $J_C^{\text{EMG}} = \lambda J_{\text{kc}}^{\text{SN}}$. Often, λ and A_n^{EMG} are both unknown. Equations with two unknown parameters are inconvenient, so these are combined into a single parameter A_f^{SN} [$= \lambda A_n^{\text{EMG}}$] called the *formal emission area for the SN barrier*.

Combining these various relations, and assuming that measured current I_m equals emission current I_e^{EMG} , yields the following EMG-theory equation:

$$I_m(F_C) = A_f^{\text{SN}} J_{\text{kc}}^{\text{SN}} = A_f^{\text{SN}} a \phi^{-1} F_C^2 \exp[-v_F b \phi^{3/2} / F_C]. \quad (2.2)$$

Although this is not explicitly shown, it needs to be understood that the values of ϕ , v_F , λ , A_n , and A_f depend on the choice of location "C".

When applying this equation to experiments, and "thinking backwards", $I_m(F_C)$ is a measured

quantity, and J_{kC}^{SN} can be calculated precisely (when ϕ and F_C are known). Thus, the *extracted* parameter $\{A_f^{SN}\}^{extr} [= I_m(F_C)/J_{kC}^{SN}]$ is, in principle, a well-defined parameter that depends on the barrier form, but not on λ : thus, the symbol $\{A_f^{SN}\}^{extr}$ carries the barrier label, rather than an equation label.

In practice, it is nearly always the *formal* area that is initially extracted from a FN or MG plot. Issues of how formal area relates to the notional area in some specific emission equation, or to geometrical quantities relating closely to real emitters, are matters for separate discussion, outside the scope of this paper. This paper is primarily about the extraction of precise values for formal area \$A_f^{SN}\$ .

A second major problem lies in determining the relationship between the characteristic barrier field F_C and the measured voltage V_m . I now prefer to write

$$F_C = V_m/\zeta_C, \quad (2.3)$$

where ζ_C is the *characteristic voltage conversion length (VCL)*, for location "C". Except in special geometries, ζ_C is not a physical length. Rather, ζ_C is a system characterization parameter: low VCL means the emitter "turns on" at a relatively low voltage V_m .

So-called ideal FE devices/systems have $I_m(V_m)$ characteristics determined *only* by the system geometry (which must be unchanging) and by the emission process, with no "complications" (see below). For ideal devices/systems, the VCL ζ_C is *constant*, and related characterisation parameters (such as characteristic FEFs) can be derived from extracted ζ_C -values (see below, and also [16]).

However, real FE devices/systems may have "complications", such as (amongst others) leakage current, series resistance in the measuring circuit, current dependence in FEFs, and space-charge effects. These may cause "non-ideality" whereby ζ_C ceases to be constant but becomes dependent on voltage and/or current. In turn, this may modify the FN or MG plot slope or cause plot non-linearity. In such cases, conventional FN-plot analysis may generate spurious results for characterization parameters [13, 16]. This will also be true for MG-plot analysis.

Additional research is urgently needed on how to analyse and model the FE $I_m(V_m)$ characteristics of non-ideal devices/systems, but it will likely be many years before comprehensive theory exists. Hence, at present, FN and MG plots provide adequate emission characterization *only* for ideal devices/systems. For FN plots there is a spreadsheet-based [18] "orthodoxy test" that can filter out non-ideal data sets; a version for MG plots will be described elsewhere in due course.

A third major problem is the following. For ideal real emitters, even if one assumes the emitter radius is large enough for the SN barrier to be an adequate approximation for evaluating tunnelling probabilities, one expects that the extracted value \${A_f^{SN}}^{extr}\$ would depend on emitter shape and on the applied voltage range. There is already material in the literature that shows that this must be the case (e.g., [19-21]). However, the FN and MG plot theories are built using "smooth planar emitter

Richard Forbes 3/10/2019 13:19

Comment [4]: Reviewer 1, Point 1.4 (part), & Point 1.6b [re: deletion below].

Richard Forbes 21/9/2019 19:59

Deleted: later, maybe in many years' time when good values for λ are known.

Richard Forbes 3/10/2019 13:19

Comment [5]: Reviewer1, Point 1.4

methodology". In this approach A_f^{SN} is treated as if it were a constant, with the extracted value $\{A_f^{\text{SN}}\}^{\text{extr}}$ derived—with varying degrees of precision—from the slope and intercept of a straight line fitted to an experimental FN or MG plot.

My current understanding is that $\{A_f^{\text{SN}}\}^{\text{extr}}$ (as derived from a FN or MG plot) is actually some kind of effective average value of $[I_m/J_{kC}^{\text{SN}}]$, taken over the range of F_C -values used in the experiments. But detailed physical interpretation of $\{A_f^{\text{SN}}\}^{\text{extr}}$ is an issue separate from whether the value extracted from a MG plot is a useful characterization parameter (which it is considered to be). Values of $\{A_f^{\text{SN}}\}^{\text{extr}}$ are presumed particularly useful for LAFEs, when comparing the properties of different emitting materials or processing regimes. Thus, having a simple method of extracting a numerically well-defined value (from a particular set of ideal experimental data) is expected to be helpful.

For LAFEs, a more useful property is perhaps the *extracted formal area efficiency* $\{\alpha_f^{\text{SN}}\}^{\text{extr}}$ (for the SN barrier), defined by

$$\{\alpha_f^{\text{SN}}\}^{\text{extr}} \equiv \{A_f^{\text{SN}}\}^{\text{extr}} / A_M, \quad (2.4)$$

where A_M is the *LAFE macroscopic area or ("footprint")*. Few experimental values have been reported for $\{a_f^{\text{SN}}\}^{\text{extr}}$. It is thought [22] to be very variable as between LAFEs, but perhaps to often lie in the vicinity of 10^{-7} to 10^{-4} . Clearly, if—for some particular LAFE material—data analysis showed (for example) that apparently only 10^{-5} % of the footprint area was actually emitting electrons, then this might indicate scope for practical improvements. This parameter looks potentially useful for technology development.

3. Theory of Murphy-Good plots

Given the above context, MG-plot theory can now be developed. This is most easily done using *scaled* parameters and equations, as follows. The *scaled (barrier) field* f (for a barrier of zero-field height ϕ) is a dimensionless physical variable formally defined, using the *Schottky constant* c_S [$\equiv (e^3/4\pi\epsilon_0)^{1/2}$] [10], by

$$f \equiv c_S^2 \phi^{-2} F_L \equiv [1.439\ 965\ \text{eV}^2\ (\text{V/nm})^{-1}] \phi^{-2} F_L. \quad (3.1)$$

For a SN barrier of zero-field height ϕ , the criterion $f=1$ defines a *reference field* F_R [$\equiv c_S^{-2} \phi^2$] at which the barrier top is pulled down to the Fermi level. For this barrier, $f=F_L/F_R$, and hence $F_L=fF_R$. It can be shown from Ref. [8] (but, better, see arXiv:1801.08251v2) that $v_f=v(x=f)$.

Scaling parameters $\eta(\phi)$ and $\theta(\phi)$ are defined by

$$\eta(\phi) \equiv bc_s^2 \phi^{-1/2}, \quad \theta(\phi) \equiv ac_s^{-4} \phi^3. \quad (3.2)$$

Substituting $F_C = f_C F_R = f_C c_s^{-2} \phi^2$ into eq. (2.2), and writing v_F explicitly as $v(f_C)$, yields the *scaled equation*

$$I_m(f_C) = A_f^{\text{SN}} \theta(\phi) f_C^2 \exp[-\eta(\phi) \cdot v(f_C) / f_C]. \quad (3.3)$$

For simplicity, we now normally cease to show the dependence of η and θ on ϕ .

The parameter f_C is helpful in characterising FE theory and the behaviour of field emitters. For example, in the case of tungsten field emitters (with $\phi=4.50$ eV) it is known that: (a) these emitters most commonly operate within the f_C -range $0.15 < f_C < 0.35$ (see supplementary-material spreadsheet related to [18]); (b) the safe operating limit for pulsed emission (in traditional field electron microscope configuration) is about $f_C < 0.6$ [23, 24], and (c) the derivation of the MG zero-temperature FE equation breaks down above about $f_C \approx 0.8$ [7]. Slightly different f_C -values would apply to materials with work-function different from 4.50 eV. Scaled-field values are easily converted back to local-field values by multiplying by the reference field F_R , which is approximately equal to 14.1 V/nm for a $\phi=4.50$ eV emitter.

A key development [25], in 2006, was the discovery of a simple good approximation for $v(f)$:

$$v_F = v(f) \approx 1 - f + (1/6) f \ln f. \quad (3.4)$$

In $0 \leq f \leq 1$, v_F takes values between $v(f=0) = 1$ and $v(f=1) = 0$. For eq. (3.4), in $0 \leq f \leq 1$, Ref. [8] found the maximum error in $v(f)$ as 0.0024 and the maximum percentage error as 0.33%. High-precision numerical formulae for $v(f)$, with maximum error 8×10^{-10} in $0 \leq f \leq 1$, are also now known (see Appendix B).

Setting $f = f_C$ and substituting eq. (3.4) into eq. (3.3) leads, after some re-arrangement, to

$$I_m(f_C) \approx A_f^{\text{SN}} \cdot \theta \exp \eta \cdot f_C^\kappa \cdot \exp[-\eta / f_C], \quad (3.5a)$$

$$\kappa \equiv 2 - \eta / 6. \quad (3.5b)$$

For an ideal device/system, eq. (3.1) can be used to define, by $V_{mR} = F_R \xi_C [= c_s^{-2} \phi^2 \xi_C]$, a *reference measured-voltage* V_{mR} at which, at location "C", the SN barrier-top is pulled down to the Fermi level. It follows that

Richard Forbes 3/10/2019 13:20
Comment [6]: Reviewer 1, Point 1.5

$$f_C = F_C/F_R = (V_m/\zeta_C) / (V_{mR}/\zeta_C) = V_m/V_{mR}, \quad (3.6)$$

and that eq. (3.5a) can be rewritten as

$$I_m(V_m) \approx \{A_f^{SN} \cdot (\theta \exp \eta) \cdot V_{mR}^{-\kappa} \cdot V_m^\kappa \cdot \exp[-\eta V_{mR}/V_m]\}, \quad (3.7)$$

and then

$$\ln\{I_m/V_m^\kappa\} \approx \ln\{A_f^{SN} \cdot (\theta \exp \eta) \cdot V_{mR}^{-\kappa}\} - \eta V_{mR}/V_m. \quad (3.8)$$

This is an equation for a *theoretical Murphy-Good plot*.

Since A_f^{SN} is being treated as constant, and all parameters on the right-hand side (except V_m) are constants, eq. (3.8) is predicted to be a *straight line* with slope S_{MG} and intercept $\ln\{R_{MG}\}$ given by:

$$R_{MG} = A_f^{SN} \cdot (\theta \exp \eta) \cdot V_{mR}^{-\kappa}, \quad (3.9)$$

$$S_{MG} = -\eta V_{mR} = -b\phi^{3/2}\zeta_C. \quad (3.10)$$

The subscript "MG" indicates that these parameters "belong to" a theoretical MG plot. It further follows that

$$R_{MG} \cdot (|S_{MG}|)^\kappa = A_f^{SN} \cdot \theta \cdot \exp \eta \cdot \eta^\kappa = A_f^{SN} \cdot \theta \eta^2 \cdot \exp \eta \cdot \eta^{-\eta/6}. \quad (3.11)$$

From equations above, $\theta \eta^2 = ab^2 \phi^2$ [$\approx (7.192492 \times 10^{-5} \text{ A nm}^{-2} \text{ eV}^{-2}) \phi^2$]. Thus, if S_{MG} and $\ln\{R_{MG}\}$ are identified with the slope S_{MG}^{fit} and intercept $\ln\{R_{MG}^{\text{fit}}\}$ of a straight line fitted to an experimental MG plot, the extracted values of the VCL ζ_C , the reference measured voltage V_{mR} , and the formal emission area A_f^{SN} are:

$$\zeta_C^{\text{extr}} = -S_{MG}^{\text{fit}} / b\phi^{3/2}, \quad (3.12)$$

$$\{V_{mR}\}^{\text{extr}} = -S_{MG}^{\text{fit}} / \eta. \quad (3.13)$$

$$\{A_f^{SN}\}^{\text{extr}} = \Lambda_{MG} \cdot R_{MG}^{\text{fit}} \cdot (|S_{MG}^{\text{fit}}|)^\kappa, \quad (3.14)$$

where the emission area extraction parameter Λ_{MG} (when using an MG plot) is given by

$$\Lambda_{\text{MG}}(\phi) \equiv 1/[(ab^2\phi^2) \cdot \exp\eta \cdot \eta^{-\eta/6}]. \quad (3.15)$$

An extracted area-efficiency value can be obtained from eqns (2.4) and (3.14), and an extracted value of macroscopic FEF γ_{M} from eq. (3.12) and the relation

$$\gamma_{\text{M}}^{\text{extr}} = d_{\text{M}} / \zeta_{\text{C}}^{\text{extr}}, \quad (3.16)$$

where d_{M} is the system distance used to define the FEF and related macroscopic field F_{M} .

Since expression (3.15) depends only on ϕ , a table of $\Lambda_{\text{MG}}(\phi)$ -values is easily prepared with a spreadsheet. Some illustrative values are shown in Table 1. $\Lambda_{\text{MG}}(\phi)$ is only weakly dependent on ϕ , so uncertainty in the true ϕ -value should cause little error in the extracted value of formal emission area.

Table 1. Typical values of quantities appearing in the "extraction formulae" (3.12) to (3.14).				
ϕ (eV)	$b\phi^{3/2}$ (V/nm)	η	$\exp\eta \cdot \eta^{-\eta/6}$	$\Lambda_{\text{MG}}(\phi)$ (nm ² /A)
2.50	27.00	6.2210	75.62	29.42
3.00	35.49	5.6790	56.55	27.32
3.50	44.73	5.2577	44.85	25.31
4.00	54.65	4.9181	37.06	23.45
4.50	65.21	4.6368	31.54	21.77
5.00	76.37	4.3989	27.46	20.25
5.50	88.11	4.1942	24.34	18.89

The consistency of the above approach has been checked by simulations that use a modified version of an already existing special-purpose spreadsheet that calculates values for the FE special mathematical functions, using the high-precision numerical expressions given in [8] and Appendix B. These MG-plot related simulations have also been compared with simulations based on the equivalent theory (set out in Appendix A) for interpreting a FN plot by using the extended MG equation. In both cases the simulations have been carried out for the characteristic-scaled-field range $0.15 \leq f_{\text{c}} \leq 0.35$, for selected values of local emitter work function in the range $2.50 \leq \phi/\text{eV} \leq 5.50$. (Emitters with $\phi=4.50$ eV are often operated within this scaled-field range.) An annotated copy of the spreadsheet as used in the simulations is provided as downloadable electronic supplementary material; details of the simulations are given in Appendix B.

In general terms, the simulations confirm that the MG plots will normally yield very consistent results for extracted values of the reference measured voltage V_{mR} , the characteristic voltage conversion length (VCL) ζ_{C} , and the formal emission area A_{f}^{SN} for the SN barrier. In these simulations,

the parameters V_{mR} and ζ_C are extracted with a consistency of 0.1% or better, and A_f^{SN} with a consistency of better than 1.8%. The corresponding figures for the FN plot are around 2% and around 52%, respectively. These results clearly demonstrate the superiority of the MG plot.

With the MG plot, there are small discrepancies between the input values for the various parameters and the "typical extracted values", as assessed by the extracted values corresponding to the scaled-field value $f_C=0.25$. These discrepancies are around 0.3% for V_{mR} and ζ_C , and up to 1.8% for A_f^{SN} , and are thought to arise because MG plot theory is based on the simple good approximation (3.4), which is not an exact expression for the function $v(f)$.

For a FN plot, as interpreted via the EMG equation, the corresponding discrepancies between the input values and "typical extracted values" are around 0.7% for V_{mR} and ζ_C , and up to 19% for A_f^{SN} . So, again, the performance of the MG plot is significantly superior to that of the FN plot.

It needs to be understood that the numerics presented here have been generated specifically for the purpose of making numerical comparisons between the performances of MG plots and FN plots, and are considered to be "validly indicative". If different values had been used for the ranges of f_C -values and/or work functions employed in the simulations, then numbers slightly different from those reported above would have been generated. However, there is no reason to think that qualitative conclusions about the comparison of MG plots and FN plots would be affected.

It is also important that one should **not** take the numerics given here as good estimates of the likely errors involved when the extraction procedures discussed here are applied to real experimental results. Further factors come into play when real experimental results are involved, including noise in the experimental data, possible uncertainty in the true work-function value, and weaknesses in the "smooth planar emitter" methodology that underlies both FN plots and MG plots. (Obviously, real emitters are very often shaped like rounded posts or pointed needles, and have atomic structure.) In the author's view, we currently have no adequate knowledge about the sizes of likely errors of this kind. The investigation of alternative $I_m(V_m)$ data interpretation methodologies and the likely errors involved are active topics of research (e.g., [19-21]).

4. Discussion

The essential merit of the Murphy-Good plot is that the whole tiresome apparatus [11,12] of slope and intercept correction factors, fitting points and chord corrections (needed for high precision parameter extraction when a FN plot is used with the EMG equation) has been swept away.

The author's view is that using MG plot analysis techniques based on the EMG equation should benefit three groups of experimentalists who currently use FN plots (and will also benefit the subject as a whole). Those who already use FN-plot interpretation theory based on eq. (3.4) will no longer

Richard Forbes 3/10/2019 13:20

Comment [7]: New material relating to simulations. Also relates to Reviewer 1, Point 1.7, and to Reviewer 2, Points 2.1 and 2.2

Richard Forbes 3/10/2019 13:20

Comment [8]: Reviewer 1, point 1.7.

Richard Forbes 3/10/2019 13:20

Comment [9]: Following text has been significantly modified as part of the response to (a) Reviewer 1, point 1.8 and (b) the introduction of Appendix A.

need to use slope and intercept correction factors, or equivalent. Those who already use the MG equation, but use formulae based ultimately on 1970s approximations for v_F , such as those of Spindt et al. [26] or Shrednik [27], will get slightly more precise results than before, and will not have to use approximation formulae whose true origin may not always be obvious.

However, the largest group of beneficiaries should be those who analyse FN plots by using the elementary FN-type equation (see [16]), which is a simplified version (see Appendix A) of the original 1928 FN FE equation, with both equations based on assuming that the tunnelling barrier is exactly triangular (ET). For this group, for *ideal* devices/systems, the simple formulae provided here allow them to precisely extract (from an MG plot) information about three characterisation parameters (the VCL, the FEF and the formal area efficiency), rather than the current normal practice of extracting only one (the FEF).

In describing these extracted results using MG plots as "precise", I refer primarily to the removal of the procedural and mathematical imprecisions associated with the use of FN plots and/or the use of 1970s era approximations for $v(f)$, s_f and r_f . There remains, of course, the possibility of physical error due to incorrect choice of emitter work function when converting experimentally determined slope and intercept values to characterisation parameters, using the extraction formulae (3.12) to (3.14). The sizes of the errors relating to particular pairs of correct and incorrect work-function values can be estimated roughly from Table 1, which shows values for the quantities that appear in these extraction formulae, for selected work-function values. More precise estimates can be obtained by using the spreadsheet: inserting a work-function value into cell K19 will generate relevant quantity values in cells K25, K31 and K 41.

One reviewer has suggested that it might be possible to overcome the above problem by applying multi-parameter numerical fitting to derive a work-function value. It is shown in the reviewer's report that this technique works effectively when applied to precisely simulated data, using the Matlab routine "fminsearch". This is an interesting suggestion that deserves to be explored further by additional simulations—but I fear that the technique may work less effectively when applied to noisy data such as may be collected in FE experiments—a point made to me by Kyritsakis (private communication, September 2019).

The following point also deserves note. Using either the original 1928 FN equation or the elementary FE equation to extract an area-like parameter from a FN plot would result in a formal-area estimate (A_f^{ET}) greater than A_f^{SN} by a factor of typically around 100 (see Appendix A). Taking the tunnelling barrier to be a SN barrier is "better physics" [5] than taking it to be the exactly triangular barrier used in deriving the elementary FE equation. Hence one expects that extracting the area A_f^{SN} should be "better scientific procedure" than extracting the area A_f^{ET} .

The formulae here envisage that researchers will use their raw $I_m(V_m)$ data to make $I_m(V_m)$ MG plots, and will then apply an orthodoxy test [18]—which must be passed if values for extracted (and related) characterisation parameters are to be regarded as trustworthy. As indicated earlier, an

Richard Forbes 3/10/2019 19:25

Comment [10]: Reviewer 2, Point 2.3c

Richard Forbes 3/10/2019 19:25

Comment [11]: Reviewer 2, Point 2.3c

Richard Forbes 26/9/2019 00:05

Comment [12]: Included for greater clarity.

orthodoxy test already exists for FN plots, and a modified version will be made available shortly for MG plots. Hopefully, this should help to reduce the incidence of spuriously high FEF values reported in the literature.

Using $I_m(V_m)$ -type MG plots could also help eliminate the widespread but unfortunate literature practice of pre-converting $I_m(V_m)$ data to become $J_M(F_M^{\text{app}})$ data before making a FN plot, where F_M^{app} is the apparent macroscopic field obtained from the pre-conversion equation, and J_M is the *macroscopic (or LAFE-average) current density* defined by $J_M = I_m/A_M$. This pre-conversion is almost always carried out by using a plausible but often *defective* conversion equation (defective because it can be invalid for non-ideal devices/systems) [13]. This in turn has often led to defective FN plots and spurious results for characterisation parameters.

Another feature of experimental FE literature is that papers sometimes use *macroscopic* current densities to show data or make FN plots, but state a formula for *local* current density in the text, without drawing attention to the difference. This practice creates un-discussed apparent discrepancies between theory and experiment, sometimes by a factor of 10^6 or more. Such confusions would be reduced if, instead, FE papers gave an equation for measured current, either an $I_m(F_c)$ equation of form (2.2) above, or a related $I_m(V_m)$ equation.

The question also arises of how improved data-analysis theory of the general kind described in this paper might be applied to non-metals, in particular semiconductors and carbon-based materials such as carbon nanotubes (CNTs). For the last 90 years or so, it has been near-universal practice amongst FE experimentalists to apply "smooth planar emitter methodology" and FN-plot theory to all materials "as a first approximation", notwithstanding that this approach was originally developed to apply to a Sommerfeld free-electron metal. The introduction of MG plots does not change this situation: MG plots can be applied to all materials "as a somewhat improved first approximation".

The problem, of course, is how to do better than this. With FN plots, it is known (certainly to the author) that differences in surface exchange-and correlation effects, as between metals and other materials, can in principle be represented by introducing new forms of slope and intercept correction factors, to replace s_i and r_i . But this is rarely if ever done. The equivalent in the present work would be to introduce a different form of data plot in which κ is taken to have a value intermediate between $(2-\eta/6)$ and 2, but good relevant theory to decide this new value of κ is not available in the literature, as far as I am aware.

A more serious difficulty, for both semiconductors and nanotubes, is the possibility of field penetration into the emitting material: this could make the operative work function ϕ^{op} depend significantly on the apex field F_a , and would require modification of the theory given here. At present, the possibility of doing this reliably is limited by the lack of good knowledge as to what the functional form of $\phi^{\text{op}}(F_a)$ would be for non-metal field emitters, in various circumstances.

It is also needful to remember that all FN and MG plots implicitly involve the (unrealistic)

Richard Forbes 26/9/2019 01:00

Comment [13]: Reviewer 3.

"smooth planar emitter" methodology. As noted earlier, the issue of how best to include emitter_shape effects, when predicting FE $I_m(V_m)$ characteristics or analysing experimental FE $I_m(V_m)$ data, is a topic of active research (e.g., [19-21]). At present, no general agreement exists on how best to perform data analysis for non-planar emitters, and significant amounts of detailed further research seem needed.

Strategically, it seems more urgent to develop $I_m(V_m)$ data interpretation theory for point-form metal emitters than to examine how to apply "smooth planar emitter" methodology to non-metals. Thus, for all the above reasons, detailed discussion of customised $I_m(V_m)$ data interpretation theory for non metals seems premature, and is outside the scope of this paper.

Development of data interpretation theory for point-form emitters will inevitably require us to eventually move on from MG plots. An early step will be to examine more general data-plot forms that might be predicted to be linear or approximately linear, in particular the so-called "power- κ " (or "power- k ") plot [21]. But, very probably, $I_m(V_m)$ data analysis will eventually find it useful or necessary to employ some more-sophisticated analysis technique, such as multi-parameter numerical fitting. This technique has been widely used outside the context of field electron emission for many years, and sometimes within it. It potentially offers greater flexibility and greater precision in parameter extraction.

The author's view is that it is likely to be some years before $I_m(V_m)$ data-interpretation methodologies specifically designed for point-form emitters (including basic theory, easy-to-use validated tools, and any related knowledge needed to interpret or use their outputs) become widely available. In particular, it would ideally need to be shown that the methodologies work robustly for "noisy" data inputs, can output "measured" values of characteristic local field and scaled field, and can provide the equivalents of an orthodoxy test [18] and (desirably) "phenomenological adjustment" [16].

Until this happens, Murphy-Good plots (which are straightforward to implement, and—like FN plots—are robust against moderate amounts of noise) can provide a significantly better approach to FE $I_m(V_m)$ data analysis than do Fowler-Nordheim plots.

Appendix A: Emission area extraction parameters for Fowler-Nordheim Plots

This Appendix gives expressions for emission area extraction parameters for Fowler-Nordheim (FN) plots. First consider the case where a FN plot is interpreted by (a) assuming that the tunnelling barrier is a SN barrier and (b) using the Extended Murphy-Good (EMG) equation. The current-voltage form of this equation is obtained by combining eqns (2.2) and (2.3) above. In natural FN coordinates this becomes

$$\ln\{I_m^{\text{EMG}}/V_m^2\} = \ln\{A_i^{\text{SN}} a \phi^{-1} \zeta_C^{-2}\} - v_F b \phi^{3/2} \zeta_C/V_m. \quad (\text{A1})$$

Richard Forbes 3/10/2019 13:30

Comment [14]: Reviewer 1, point 1.9.

Richard Forbes 3/10/2019 13:30

Comment [15]: Reviewer 1, point 1.6a

Richard Forbes 12/9/2019 03:05

Deleted: , impracticable to summarize here

Richard Forbes 11/9/2019 11:14

Deleted: consensus

Richard Forbes 26/9/2019 01:01

Comment [16]: Reviewer 3

Richard Forbes 23/9/2019 06:57

Deleted: take several years

Richard Forbes 11/9/2019 12:54

Deleted: or more to reach consensus, and many further years to develop fully correct theory. Expectation is that in due course we shall need to move on from MG plots

Richard Forbes 3/10/2019 21:33

Comment [17]: Reviewer 1, Points 6c & 6d.

Richard Forbes 25/9/2019 07:48

Comment [18]: Appendix partly introduced in response to Reviewer 1, Point 8. But also because it seems a good idea to put this piece of theory into the literature anyhow.

Customising the general theory in Ref. [11] yields the slope S^{\tan} of the tangent to the "theoretical" plot (A1) as

$$S^{\tan}(V_m^{-1}) = -s(f_C) \cdot b \phi^{3/2} \xi_C, \quad (\text{A2})$$

where $f_C [= V_m/V_{mR}]$ is the characteristic scaled-field value corresponding to measured voltage V_m , V_{mR} is the reference measured voltage as discussed in the main text, and $s(f)$ is the slope correction function for the SN barrier, as usually defined (e.g., [8]). Also, from [11], the intercept $\ln\{R^{\tan}\}$ that this tangent makes with the vertical ($1/V_m = 0$) axis is given via

$$R^{\tan}(V_m^{-1}) = r(f_C) \cdot A_F^{\text{SN}} a \phi^{-1} \xi_C^{-2}, \quad (\text{A3})$$

where $r(f_C)$ is the 2012 intercept correction function as defined in [11] and denoted there by r_{2012} .

Because a theoretical FN plot of the EMG equation is slightly curved, its slope (and hence the slope of its tangent) vary with the horizontal-axis coordinate V_m^{-1} . The *tangent method* of plot interpretation takes a given experimental FN plot to be parallel to this theoretical tangent as defined at a particular V_m^{-1} -value and hence at a particular f_C -value f_i . *Fitting values* of the correction functions are then defined by $s_i = s(f_i)$ and $r_i = r(f_i)$. On identifying the related values of S^{\tan} and $\ln\{R^{\tan}\}$ with the slope $S_{\text{FN}}^{\text{fit}}$ and intercept $\ln\{R_{\text{FN}}^{\text{fit}}\}$ of the straight line fitted to the experimental FN plot, we find that

$$R_{\text{FN}}^{\text{fit}} | S_{\text{FN}}^{\text{fit}}|^2 = (r_i s_i^2) (ab^2 \phi^2) A_F^{\text{SN}} \quad (\text{A4})$$

Hence, the extracted value of A_F^{SN} is given in terms of $S_{\text{FN}}^{\text{fit}}$ and $R_{\text{FN}}^{\text{fit}}$ by the extraction equation

$$\{A_F^{\text{SN}}\}^{\text{extr}} = A_{\text{FN}}^{\text{SN}} \cdot (R_{\text{FN}}^{\text{fit}} | S_{\text{FN}}^{\text{fit}}|^2) \quad (\text{A5})$$

where the extraction parameter $A_{\text{FN}}^{\text{SN}}$ for a FN plot, interpreted by assuming a SN barrier, is given by

$$A_{\text{FN}}^{\text{SN}} = 1 / [(ab^2 \phi^2) (r_i s_i^2)]. \quad (\text{A6})$$

The fitting value f_i is not initially known. In principle, it can be estimated by an iterative process, but normal practice takes $s_i = 0.95$ as a first approximation. This corresponds to $f_i \cong 0.2815$ and (for an emitter with work-function 4.500 eV) to $r_i s_i^2 \cong 112.9$. The corresponding extraction-parameter value is

$$A_{\text{FN}}^{\text{SN}} \sim 6.083 \text{ nm}^2 / \text{A} \quad (\text{A7})$$

If, instead, a FN plot is interpreted by assuming the tunnelling barrier is exactly triangular (ET), then a numerically different result is found for the related extraction parameter $A_{\text{FN}}^{\text{ET}}$. In this case an "extended elementary (EEL) equation" [16] is written in the current-voltage form

$$I^{\text{EEL}}(V_m) = A_{\text{ET}}^{\text{ET}} a \phi^{-1} \zeta_{\text{C}}^{-2} V_m^2 \exp[-b \phi^{3/2} \zeta_{\text{C}} / V_m], \quad (\text{A8})$$

where $A_{\text{ET}}^{\text{ET}}$ is the formal emission area for the ET barrier. The extracted value of $A_{\text{ET}}^{\text{ET}}$ is given in terms of $S_{\text{FN}}^{\text{fit}}$ and $R_{\text{FN}}^{\text{fit}}$ by

$$\{A_{\text{ET}}^{\text{ET}}\}^{\text{extr}} = A_{\text{FN}}^{\text{ET}} \cdot (R_{\text{FN}}^{\text{fit}} | S_{\text{FN}}^{\text{fit}}|^2), \quad (\text{A9})$$

where

$$A_{\text{FN}}^{\text{ET}} = 1 / (ab^2 \phi^2). \quad (\text{A10})$$

This result is found from eq. (A6) by noting that, for the ET barrier, r_{t} and s_{t} are both replaced by unity. For $\phi = 4.500 \text{ eV}$, eq. (A10) yields

$$A_{\text{FN}}^{\text{ET}} \sim 686.6 \text{ nm}^2 / \text{A} \quad (\text{A11})$$

To achieve numerical consistency in making comparisons, values (A7) and (A11) are given here to four significant figures, but the physical precision is very much worse, particularly for value (A7), which could easily be in error by 10% or more.

Clearly, for a given value of the experimentally derived product $(R_{\text{FN}}^{\text{fit}} | S_{\text{FN}}^{\text{fit}}|^2)$, use of the extraction-parameter value (A11) will lead to estimates of the formal emission area $A_{\text{ET}}^{\text{ET}}$ that are much larger (by a factor of order 100) than those found by using the extraction-parameter value (A7) to estimate the formal emission area $A_{\text{ET}}^{\text{SN}}$. Qualitatively, this is not surprising, since it is known (e.g., [5]) that the 1956 MG FE equation predicts emission current densities that are larger than those predicted by the elementary FE equation, by a factor typically between 250 and 500. This result underlines the need for careful definition of area-like quantities.

More important is the following conclusion. As shown in Appendix B, extracted values of $A_{\text{ET}}^{\text{SN}}$ found by analysing a MG plot are much the same as the extracted values of $A_{\text{ET}}^{\text{SN}}$ found by using the extended MG equation to analyse a FN plot. This means that extracted values of $A_{\text{ET}}^{\text{SN}}$ found by

analysing a MG plot are much smaller (by a factor of order 100) than extracted values of A_f^{ET} found by using the extended elementary equation to analyse a FN plot. Both these analysis procedures are relatively straightforward. However, when one accepts (for reasons discussed in [5]) that assuming a SN barrier is better physics than assuming an ET barrier, then the conclusion is that A_f^{SN} is physically a "more meaningful parameter" than A_f^{ET} , and that extracting an A_f^{SN} -value rather than an A_f^{ET} -value is "significantly better scientific procedure".

Appendix B: Description and discussion of simulation procedures and results

This Appendix describes simulations carried out in order to test the methodology proposed in this paper for extracting A_f^{SN} values from a Murphy-Good plot, and to compare the precision of the methodology with that of the corresponding procedure for extracting A_f^{SN} values from a Fowler-Nordheim plot. For simplicity, these simulations make use of an already existing special-purpose spreadsheet able to evaluate high-precision values of the FE special mathematical functions $v(x)$ and $u(x)$ [$\equiv -dv/dx$] (and hence of all the FE special mathematical functions, and of related quantities such as emission current densities). The parameter x is the *Gauss variable* (i.e., the independent variable in the Gauss Hypergeometric Differential Equation). These two functions are estimated by the following series, derived from those given in [8] by replacing the symbol l' by the symbol x now preferred, and by slightly adjusting the form of the resulting series for $v(x)$ (without changing its numerical predictions):

$$v(x) \equiv (1-x) \left(1 + \sum_{i=1}^4 p_i x^i \right) + x \ln x \sum_{i=1}^4 q_i x^{i-1} \quad (B1)$$

$$u(x) \equiv u_1 - (1-x) \sum_{i=0}^5 s_i x^i - \ln x \sum_{i=0}^4 t_i x^i \quad (B2)$$

Values of the constant coefficients p_i , q_i , s_i and t_i are shown in Table 3.

Table 3. Numerical constants for use in connection with eqs (B1) and (B2).

i	p_i	q_i	s_i	t_i
0	-	-	0.053 249 972 7	0.187 5
1	0.032 705 304 46	0.187 499 344 1	0.024 222 259 59	0.035 155 558 74
2	0.009 157 798 739	0.017 506 369 47	0.015 122 059 58	0.019 127 526 80
3	0.002 644 272 807	0.005 527 069 444	0.007 550 739 834	0.011 522 840 09
4	0.000 089 871 738 11	0.001 023 904 180	0.000 639 172 865 9	0.003 624 569 427
5	-	-	-0.000 048 819 745 89	-

$u_1 = 3\pi/8\sqrt{2} \approx 0.8330405509$

Richard Forbes 25/9/2019 08:55

Comment [19]: Additional material, partly stimulated by Reviewer 2, Point 2.1.

I am also inclined to think that describing these simulation results in detail (and providing the related spreadsheet as supplementary material) aligns better with Royal Society policies relating to the transparency of numerical calculations.

It is readily seen that, at the values $x=0,1$, eq. (B1) generates the exactly correct values $v(0)=1$, $v(1)=0$, and that at $x=1$, eq. (B2) generates the exactly correct value $u(1) = u_1 = 3\pi/8\sqrt{2}$.

The form of eq. (B1) mimics the form of the lower-order terms in the (infinite) exact series expansion for $v(x)$ [6], but the coefficients in Table 3 have been determined by numerical fitting to exact expressions for $v(x)$ and $u(x)$ (in term of complete elliptic integrals) evaluated by the computer algebra package MAPLE™. In the range $0 \leq x \leq 1$ (but not outside it), $v(x)$ takes values lying in the range $1 \geq v(x) \geq 0$, and the maximum error associated with formulae (B1) and (B2) is known to be less than 8×10^{-10} [8]. The accuracy of the spreadsheet implementation is expected to be similar [e.g., see Wikipedia entry on "Numeric precision in Microsoft Excel"].

These formulae are applied in the context of Murphy-Good-type FE equations by setting $x=f_C$. A copy of the modified spreadsheet, as used in the present simulations, is provided as electronic supplementary material and will need to be downloaded.

For these simulations, the FE device/system has been taken as ideal, the local work function ϕ has been taken as 4.50 eV, the input value of the SN-barrier formal emission area A_f^{SN} has been taken as constant and equal to 100 nm^2 , and the input value of the reference measured voltage V_{mR} has been taken as constant and equal to 6000 V. For a work-function value of 4.500 eV, this V_{mR} value is equivalent to a constant characteristic voltage conversion length ζ_C of approximately 426.66 nm.

It is known (see spreadsheet in electronic supplementary material related to [18]) that tungsten field emitters (with assumed work function 4.50 eV) normally operate within the range $0.15 \leq f_C \leq 0.35$. In this range, for f_C -values increasing by steps of 0.01, values have been calculated (in the spreadsheet related to the present paper) for the measured voltage V_m (column AM), its reciprocal V_m^{-1} (column AQ), the characteristic kernel current density J_{kC}^{SN} (column AN), the predicted measured current J_m^{EMG} (column AO), and the MG-plot vertical-axis quantity $\ln\{J_m^{EMG} / V_m^k\}$ (column AR).

For each of the f_C values in the range $0.20 \leq f_C \leq 0.30$, an "extracted local slope" S_{MG} has been estimated (column AS) by using the equation

$$S_{MG} \approx \{Y(f_C-0.05) - Y(f_C+0.05)\} / \{X(f_C-0.05) - X(f_C+0.05)\}, \quad (B3)$$

where $X [= 1/V_m]$ and $Y [= \ln\{J_m^{EMG} / V_m^k\}]$ are the quantities on the horizontal and vertical axes of the MG plot. The parameter S_{MG} given by eq. (B3) is the average slope over a scaled-field range of 0.1, centred on the chosen f_C -value. S_{MG} is then used to derive an estimate (column AT) for the vertical-axis ($1/V_m=0$) intercept $\ln\{R_{MG}\}$ of the tangent to the MG plot at the chosen f_C -value, using a formula equivalent to

$$\ln\{R_{MG}\} \approx Y(f_C) + |S_{MG}| X(f_C). \quad (B4)$$

In order to make comparisons with extraction procedures that use a FN plot (as interpreted using the EMG equation) to estimate a value for A_f^{SN} , we have carried out manipulations similar to those just described, but with κ taken equal to exactly 2. Columns BF and BG show the resulting values of S_{FN}^{SN} and $\ln\{R_{FN}^{SN}\}$.

In relation to slope and intercept values extracted from the simulations, the observed near-constancy of the values in columns AS and AT shows that the MG plot is "almost exactly" straight. The plot is not expected to be exactly straight, because MG plot theory is based on the "simple good approximation" (3.4), which is not an exactly correct formula for $v(f_C)$.

Over the range of midpoint f_C -values considered, namely $0.20 \leq f_C \leq 0.30$, for the MG plot the variation in the extracted local slope is about 0.06 % and that in the extracted intercept is about -0.1 %. The corresponding figures for the FN plot are about 1.9 % and about -2.0 %. This confirms that the MG plot is much more closely linear than the FN plot.

For the parameters $\{V_{mR}\}^{extr}$ and ζ_C^{extr} that can be derived from the extracted slope, the derived variations are, of course, the same as the variations in the extracted slope. However, comparisons can also be made between the input value (6000 V for V_{mR}) and the extracted value $\{V_{mR}\}^{extr}$ for the central f_C -value in the whole range considered. For this value ($f_C=0.25$), $\{V_{mR}\}^{extr}$ is 5982.5 V for the MG plot, 6041.6 V for the FN plot. These values quantify discrepancies between the input and extracted values of V_{mR} : for the MG plot the discrepancy is -0.29%, for the FN plot the discrepancy is +0.69%. For the MG plot, the discrepancy is probably caused by the use of the "simple good approximation" to develop MG plot theory. The same figures and thinking apply to the extraction of characteristic VCL values, and to the extraction of characteristic FEF values via eq. (3.16).

For the parameter $\{A_f^{SN}\}^{extr}$ extracted using an MG plot and eqs (3.13.) and (3.14), the variation in this parameter over the mid-point range is about 1.3 %, and the discrepancy between the input value and the central extracted value is about -1.4 %. When this parameter is extracted using a FN plot and eqs (A5) and (A6), the variation over the midpoint range is about 40% and the discrepancy between the input value and the central extracted value is about 15%. These figures confirm that, for the purpose of extracting a precise estimate of A_f^{SN} , the MG plot is demonstrably much superior to the FN plot.

The numerics presented here have been derived primarily for the purpose of comparing the merits of FN plots and MG plots. As noted in the main text, those for the MG plot should not be taken as good estimates of the likely errors involved in extracting characterization-parameter values from real experimental data.

Acknowledgments

Research by Dr Eugeni O. Popov and colleagues at the Ioffe Institute in Saint-Petersburg have been a major stimulus for this work. Their numerical simulations (e.g., Ref. [28]) relating to carbon nanotubes have looked for the value of k in the empirical FE equation $I_m = CV_m^k \exp[-B/V_m]$ proposed [29] some years ago. My thinking about how to find C -values has led to this article. I thank Dr Popov for numerous e-mail discussions.

References

1. Stern TE, Gossling BS, Fowler RH. 1929 Further studies in the emission of electrons from cold metals. *Proc. R. Soc. Lond. A* **124**, 699-723.
2. Fowler RH, Nordheim L. 1928 Electron emission in intense electric fields. *Proc. R. Soc. Lond. A* **119**, 173-181.
3. Burgess RE, Kroemer H, Houston JM. 1953 Corrected values of Fowler-Nordheim field emission functions $v(y)$ and $s(y)$. *Phys. Rev.* **90**, 515.
4. Nordheim LW. 1928 The effect of the image force on the emission and reflection of electrons by metals. *Proc. R. Soc. Lond. A* **121**, 626-639.
5. Forbes RG. 2019 Comments on the continuing widespread and unnecessary use of a defective emission equation in field emission related literature, arXiv:1906.10277v3
6. Deane JHB, Forbes RG. 2008 The formal derivation of an exact series expansion for the Principal Schottky-Nordheim Barrier Function v , using the Gauss Hypergeometric Differential Equation. *J. Phys. A: Math. Theor.* **41**, 395301.
7. Murphy EL, Good RH. 1956 Thermionic emission, field emission and the transition region. *Phys. Rev.* **102**, 1464-1473.
8. Forbes RG, Deane JHB. 2007 Reformulation of the standard theory of Fowler-Nordheim tunnelling and cold field electron emission. *Proc. R. Soc. Lond. A* **463**, 2907-2927.
9. *International Standard ISO 80000-1:2009. Quantities and units—Part 1 General*, (ISO, Geneva).
10. Forbes RG, Deane JHB. 2011 Transmission coefficients for the exact triangular barrier: an exact general analytical theory that can replace Fowler & Nordheim's 1928 theory. *Proc. R. Soc. Lond.*

A **467**, 2927-2947. See Electronic Supplementary Material for information about special universal constants used in field emission.

11. Forbes RG, Fischer A, Mousa MS. 2013 Improved approach to Fowler-Nordheim plot analysis. *J. Vac. Sci. Technol. B* **31**, 02B103.
12. Forbes RG, Deane JHB. 2017 Refinement of the extraction-parameter approach for deriving formal emission area from a Fowler-Nordheim plot. 30th International Vacuum Nanoelectronics Conf., Regensburg, July 2017. [Technical Digest, pp. 234-235.] (doi:10.13140/RG.2.2.33297.74083)
13. Forbes RG 2019 Why converting field emission voltages to macroscopic fields before making a Fowler-Nordheim plot has often led to spurious characterization results, J. Vac Sci. Technol. B 37, 051802. (doi: 10.1116/1.5111455)
14. Modinos A. 2001 Theoretical analysis of field emission data. *Solid-State Electronics* **45**, 809-816.
15. Lepetit B. 2017 Electronic field emission models beyond the Fowler-Nordheim one. *J. Appl. Phys.* **122**, 215105. (doi:10.1063/1.5009064)
16. Forbes RG, Deane JHB, Fischer A, Mousa MS. 2015 Fowler-Nordheim plot analysis: a progress report. *Jordan J. Phys.* **8**, 125-147. (Also see: arXiv:1504.01634v7.)
17. Forbes RG. 2018 Comparison of the Lepetit field emission current-density calculations with the Modinos-Forbes uncertainty limits. 31st International Vacuum Nanoelectronics Conf., Kyoto, July 2018. [Technical Digest, pp. 126-127] (doi:10.13140/RG.2.2.35893.73440/1)
18. Forbes RG. 2013 Development of a simple quantitative test for lack of field emission orthodoxy. *Proc. R. Soc. Lond. A* **469**, 20130271. (doi:10.1098/rspa.2013.027)
19. Jensen KL. 2018 Introduction to the Physics of Electron Emission (Wiley, Chichester, UK).
20. Biswas D, Kumar R. 2019 Validation of current formula for a metallic nanotipped field emitter. J. Vac. Sci. Technol. B 37, 040603.
21. Forbes RG, Deane JHB, Kolosko AG, Fillipov SV, Popov EO. 2019 Reinvigorating our approach to field emission area extraction (because Murphy-Good plots are better than Fowler-Nordheim plots). 32nd International Vacuum Nanoelectronics Conf. & 12th International Vacuum Electron Sources Conf., Cincinnati, July 2019. [Technical Digest, p. 23.] (doi:10.13140/RG.2.2.32112.81927)
22. Forbes RG. 2009 Use of the concept "area efficiency of emission" in equations describing field emission from large area electron sources. *J. Vac. Sci. Technol. B* **27**, 1200-1203. (doi: 10.1116/1.3137964)
23. Dyke W, Trolan JK. 1953 Field emission: large current densities, space charge, and the vacuum arc. 1953 Phys. Rev. 89, 799-808.
24. Forbes RG. 2008 Description of field emission current/voltage characteristics in terms of scaled barrier field values (\$f\$ -values). J. Vac. Sci. Technol. B 26, 209-213.

25. Forbes RG. 2006 Simple good approximations for the special elliptic functions in standard Fowler-Nordheim tunnelling theory for a Schottky-Nordheim Barrier. *Appl. Phys. Lett.* **89**, 113122. (doi:10.1063/1.2354582)
26. Spindt CA, Brodie I, Humphrey L, Westerberg ER. 1976 Physical properties of thin-film field emission cathodes. *J. Appl. Phys.* **47**, 5248-5263.
27. Shrednik VN. 1974 "Theory of field emission", Chap. 6 in: Elinson MI (ed.) "*Unheated Cathodes*" ("Soviet Radio", Moscow, 1974) (In Russian). See eq. 6.10.
28. Popov EO, Kolosko AG, Filippov SV. 2018 Experimental definition of k -power of pre-exponential voltage factor for LAFE. 2018 31st International Vacuum Nanoelectronics Conf., Kyoto, July 2018 (IEEE, Piscataway, 2018), [Technical Digest, pp. 254-255.]
29. Forbes RG. 2008 Call for experimental test of a revised mathematical form for empirical field emission current-voltage characteristics. *Appl. Phys. Lett.* **92**, 193105. (doi:10.1063/1.2918446)